
# Flexible forecast value metric suitable for a wide range of decisions: application using probabilistic subseasonal streamflow forecasts

Richard Laugesen[1,2], Mark Thyer[1], David McInerney[1], Dmitri Kavetski[1]

[1]School of Civil, Environmental and Mining Engineering, University of Adelaide, SA, Australia
[2]Bureau of Meteorology, Canberra, ACT, Australia

*Correspondence to:* Richard Laugesen (richard.laugesen@bom.gov.au)

**Abstract.** Forecasts have the potential to improve decision-making but have not been widely evaluated because current
forecast value methods have critical limitations. The ubiquitous Relative Economic Value (REV) metric is limited to binary
decisions, cost-loss economic model, and risk neutral decision-makers. Expected Utility Theory can flexibly model more
real-world decisions, but its application in forecasting has been limited and the findings are difficult to compare with those
from REV. A new metric, Relative Utility Value (RUV), is developed using Expected Utility Theory. RUV has the same
interpretation as REV which enables a systematic comparison of results, but RUV is more flexible and able to handle a wider
range of real-world decisions because all aspects of the decision-context are user-defined. In addition, when specific
assumptions are imposed it is shown that REV and RUV are equivalent. We demonstrate the key differences and similarities
between the methods with a case study using probabilistic subseasonal streamflow forecasts in a catchment in the southern
Murray-Darling Basin of Australia. The ensemble forecasts were more valuable than a reference climatology for all lead-
times (max 30 days), decision types (binary, multi-categorical, and continuous-flow), and levels of risk aversion for most
decision-makers. Beyond the second week however, decision-makers who were highly exposed to damages should use the
reference climatology for the binary decision, and forecasts for the multi-categorical and continuous-flow decision. Risk
aversion impact was governed by the relationship between the decision thresholds and the damage function, leading to a
mixed impact across the different decision-types. The generality of RUV makes it applicable to any domain where forecast
information is used for making decisions, and the flexibility enables forecast assessment tailored to specific decisions and
decision-makers. It complements forecast verification and enables assessment of forecast systems through the lens of
customer impact.

## 1    Introduction

Effective water resource management is critically important to human welfare, thriving environmental ecosystems,
agricultural productivity, power generation, town supply and economic growth (United Nations, 2011; UNESCO, 2012).
These decisions depend primarily on the current and anticipated hydrometeorological conditions and are frequently informed



by forecasts. In particular, many decisions, such as reservoir operations and early flood warnings, appear to benefit from forecasts at a subseasonal time horizon (2-8 week lead-times) because of long river travel times, operational constraints, and logistical overheads (White et al., 2015; Monhart et al., 2019; Schmitt Quedi and Mainardi Fan, 2020; McInerney et al., 2020). These studies use forecast verification techniques and demonstrate that subseasonal streamflow forecasts are becoming more skilful at longer lead-times with reliable estimates of uncertainty. However, it is not clear whether forecasts should be used to inform water-sensitive decisions once economic and other factors are considered, the key question is, do the forecasts provide value for the decisions-makers. These factors are typically not considered when evaluating the performance of forecasts using forecast verification. This study aims to address this gap by quantifying the value of subseasonal streamflow forecasts for water-sensitive decisions, such as storage release management and environmental watering.

Forecast verification is the comparison of a set of forecasts spanning a historical period to the observed record using statistical performance metrics. The hydrological forecasting community uses numerous statistical metrics to summarise the performance of ensemble forecasts, including the Continuous Rank Probability Score (CRPS) for accuracy and metrics based on the Probability Integral Transform for statistical reliability for example (Cloke and Pappenberger, 2009; McInerney et al., 2017; Woldemeskel et al., 2018; Bennett et al., 2021). Forecast verification is essential but insufficient for decision-makers to confidently adopt forecasts into their operational and strategic decision-making processes. It does not consider the broader context a decision is made in, the economic trade-offs and different decision types for example. Forecast value measures how much better a decision is when made using one source of forecast information relative to another. It explicitly considers the broader decision context, economics being one of the most tractable aspects to analyse. When using forecast verification metrics as a proxy for forecast value we are implicitly assuming that better verification implies more value. However, additional skill is not necessarily a good predictor of additional benefit to a decision-maker (Murphy, 1993; Roebber and Bosart, 1996; Marzban, 2012).

In this paper we consider the value of streamflow forecasts to improve the outcome of binary, multi-categorical, and continuous-flow decisions and require a method to quantify this value. However, the most frequently used forecast value method in hydrology and meteorology is Relative Economic Value (REV) which is unable to handle a wide range of decision-types. Substantial research in the field of meteorology has explored the value of temperature, wind and rainfall forecasts for user decisions using REV (e.g. Richardson, 2000; Wilks, 2001; Mylne, 2002; Palmer, 2002; Zhu et al., 2002; Foley and Loveday, 2020; Dorrington et al., 2020). There is an ongoing interest in hydrology to quantify the value of forecasts for decision-making using REV (e.g. Laio and Tamea, 2007; Roulin, 2007; Bergh and Roulin, 2010; Weijs et al., 2010; Verkade and Werner, 2011; Bogner et al., 2012; Abaza et al., 2013; Fundel et al., 2013; Abaza et al., 2014; Thiboult et al., 2017; Verkade et al., 2017; Portele et al., 2021) but no application for subseasonal streamflow forecasts. REV is convenient in its tractability but has strong assumptions about the decision type, economic model, and decision-maker behaviour which neglects important aspects of decision-making and which have implications on the conclusions reached (Tversky and Kahneman, 1992; Katz and Murphy, 1997; Matte et al., 2017).



REV is only suitable to assess forecast value for a limited set of decisions; binary categorical, cost-loss economic model, event frequency as a reference baseline forecast, and risk-neutral decision-makers (Thompson, 1952; Murphy, 1977). This limited setup is an excellent prototypical decision model which is useful to understand the salient features of forecast value but may give misleading results when used to model real-world decisions. For example, flood warnings are a practically important multi-categorical decision, typically classified into either minor, moderate, or major flood impact levels, whereas

REV only handles binary decisions. Likewise, adjusting the release of water from a storage is best informed by continuous-flow forecasts and may require a more complex economic model than the cost-loss economic model assumed by REV. REV is unable to consider the impact of risk-averse decision-makers; a preference for future options with more certainty, even though it may lead to a less economically beneficial outcome. For example, a water authority deciding to announce a large water allocation event or an irrigator placing an order for water may prefer forecasts with a more certain outcome (high risk

aversion), even if it means missing out on some additional economic benefit. Conversely, a storage operator deciding whether to stop a release because they are concerned about flood damage downstream may exhibit low aversion to risk and tolerate uncertain forecasts of downstream tributary inflows because the economic loss would be significant if a release coincided with a high flow tributary event.

The field of decision theory explores how agents make decisions with uncertain information and has produced a number of

innovations, such as Expected Utility Theory (Neumann, 1944; Mas-Colell, 1995). Expected Utility Theory is flexible enough to model different decision types, economic models, and risk aversion but there is limited understanding of the relationship and differences between it and REV. It proposes that when faced with a choice a rational person will select the option leading to an outcome that maximises their utility; an ordinal measure based on the ranking of outcomes. Different people may rank outcomes differently because of their specific preferences, such as risk aversion. While Expected Utility

Theory is widely used in economics, public policy, and financial management, it has had a very limited application in hydrology and associated fields. Matte et al. (2017) recently used it to assess the impact of increasing intangible losses and risk aversion on the value of raw probabilistic streamflow forecasts for a single multi-categorical decision type with 12 flow classes. Although the foundation method is general the application was case study specific, limited to a single multi-categorical decision, used metrics which are somewhat unfamiliar to the verification community. The results were not

presented on a traditional Value Diagram and therefore no comparison to REV could be made. The authors are unaware of any literature which attempts to align REV with forecast value from Expected Utility Theory or present the results on a Value Diagram. There is no method available to the verification community to flexibly evaluate the value of probabilistic forecasts for different decision types, economic models, or decision-makers (Cloke and Pappenberger, 2009; Soares et al., 2018).

Probabilistic forecasts of continuous hydrometeorological variables lead to improved forecast verification in many cases and are operationally delivered by all major forecast producers but decision-makers are still learning the most effective way to use them (Duan et al., 2019; Carr et al., 2021). A common approach for decision-makers to use probabilistic forecasts is to first converted them to deterministic forecasts using a fixed critical probability threshold (Fundel et al., 2019; Wu et al.,





2020). This approach is known to lead to sub-optimal forecast value in some situations through studies using REV
(Richardson, 2000; Wilks, 2001; Zhu et al., 2002; Roulin, 2007). Matte et al. (2017) quantified forecast value with an
alternative decision making approach which uses the whole forecast distribution to decide on an ideal action at each forecast
update. It is not clear that this alternative approach leads to better decision outcomes and the authors are unaware of any
literature comparing them.

This study aims to:

1.    Develop a methodology to systematically compare two forecast value techniques; REV and a method based on
            Expected Utility Theory.

       2.    Demonstrate the key differences and similarities between the approaches for different decision types and levels of
            risk aversion using subseasonal streamflow forecasts in the Murray-Darling Basin.

In Sect. 2 , the theoretical background of REV and an Expected Utility Theory approach for forecast value are introduced.
Section 3  proposes a new metric (Relative Utility Value) based on Expected Utility Theory and details its equivalence to
REV when a set of assumptions are imposed. The methodology for a case study using subseasonal forecasts with binary,
multi-categorical, and continuous-flow decisions is introduced in Sect. 4 . Results of the case study are presented in Sect. 5
and discussed in Sect. 6 , including implications for forecast users and producers. Conclusions are drawn in Sect. 7 .

## 2        Theoretical background

The background theory introduced here focuses on two methods to quantify the value of forecasts, REV and an approach
using Expected Utility Theory introduced by Matte et al. (2017).

### 2.1      Relative economic value

REV is a frequently used and excellent method to quantify the value of forecasts for cost-loss binary decision problems
(Richardson, 2000; Wilks, 2001; Zhu et al., 2002). Cost-loss is a well-studied economic model where some of the loss due to
a future event can be avoided by deciding to pay for an action which will mitigate it (Thompson, 1952; Murphy, 1977; Katz
and Murphy, 1997). Many real-world decisions, such as insurance, can be simplified and framed in this way as a binary
categorical decision. The method assumes that any real-world decision it is applied to can be framed in this way.

### 2.1.1    REV with deterministic forecasts

Whether a user is expected to benefit in the long run from the use of a forecast system (or an alternative) can be assessed
using a 4x4 contingency table. Table 1 includes the hit rate $h$ , miss rate $m$ , false alarm rate $f$ , and correct rejection rate
(quiets) $q$ from a long run historical simulation, along with the net expense from each outcome where $C$ is the cost of an





action to mitigate the loss $L$. However, only a portion $L_a$ of the total loss can be avoided with the remainder $L_u$ being unavoidable.

**Table 1: Contingency table for the cost-loss decision problem with expenses from each possible outcome. Here $C$ is the cost of the mitigating action, $L_u$ is the unavoidable portion of loss $L$ from the event occurring, and $L_a$ is avoidable portion of loss from the action.**

|  | Event occurred | Event did not occur |
|---|---|---|
| Action taken | Hit rate ($h$) <br> $C + L_u$ | False alarm rate ($f$) <br> $C$ |
| Action not taken | Miss rate ($m$) <br> $L = L_a + L_u$ | Quiets/correct rejection rate ($q$) <br> 0 |

The expected long run expense $E$ of each outcome depends on the rate that outcome occurred over some historical period, and these rates will be different depending on which forecast information is used. The REV metric is constructed by comparing the relative difference in the total net expenses for decisions made using forecast, perfect, and climatological baseline information.

$$V = \frac{E_{\text{climate}} - E_{\text{forecast}}}{E_{\text{climate}} - E_{\text{perfect}}} \tag{1}$$

where each expense term is the summation of the contingency table elements each weighted by the rate of occurrence. Equation (1) is equivalent to the following standard analytical equation for REV (Zhu et al., 2002) when the long run average expenses from Table 1 are considered.

$$V = \frac{\min(\bar{o}, \alpha) - (h + f)\alpha - m}{\min(\bar{o}, \alpha) - \bar{o}\alpha} \tag{2}$$

Where the parameter $\alpha$ is known as the cost-loss ratio.

$$\alpha = \frac{C}{L_a} \tag{3}$$

The derivation of Eq. (2) is available in the Supplement. Equation (2) is typically applied over a range of $\alpha$ values and this set of REV results is plotted on a Value Diagram. This diagram provides a visualisation of how forecast value varies for decision-makers with different exposure to losses, and by extension exposure to the underlying damages. We can consider users with a small cost-loss ratio to have a smaller exposure to damages due to their enhanced ability to leverage a small amount of spending (small cost) to avoid larger future damages (large loss). Conversely, users with a large cost-loss ratio





would have a larger exposure to damages. For the same event the level of exposure will vary for different decision-makers

and decision types.

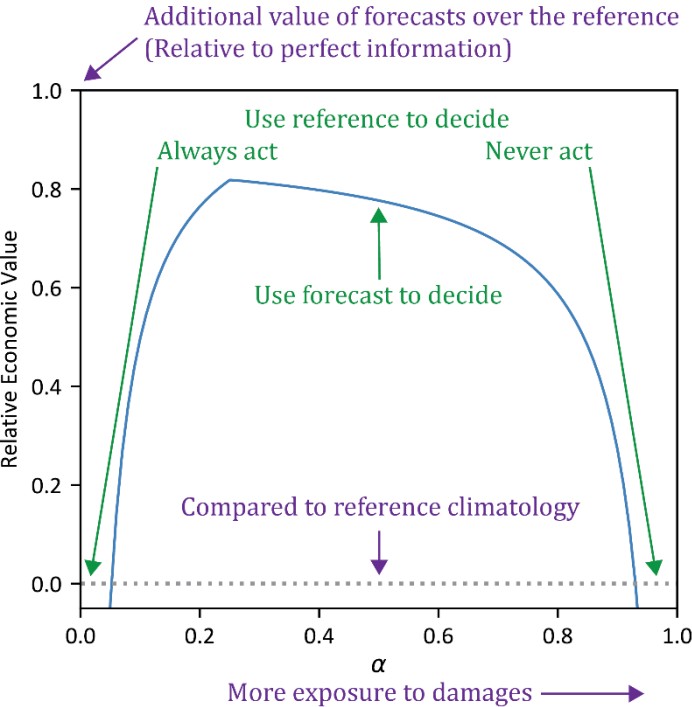

**Figure 1: Illustrative value diagram with key features annotated.**

Figure 1 presents an illustrative Value Diagram as an aid to describe its interpretation. The non-dimensional cost-loss ratio $\alpha$ is shown on the x-axis and can be interpreted as a continuum of different decision-makers using the forecasts, with increasing exposure to the damages. A value of $\alpha = 1$ corresponds to maximum exposure: if losses are $100,000 then the amount to spend on a mitigating action is also $100,000. A value of $\alpha = 0.1$ indicates that only $10,000 would be needed to mitigate the loss. The y-axis shows forecast value according to REV and has a similar interpretation to any skill-score based

metric. A value of REV = 1 indicates that decisions made using forecast information successfully mitigated the same level of losses (over the historical period) as decisions made using perfect information (streamflow observations). A value of REV = 0 indicates the decisions were only as good as those made using reference forecast information (climatology). A negative value indicates the decisions were worse than the reference. A value of REV = 0.7, for example, indicates that on average the decisions made using forecasts would have successfully mitigated 70% more losses over the historical period

than decisions made using the reference forecast.




### 2.1.2 REV with probabilistic forecasts

Constructing a value diagram using Eq. (2) is only possible with categorical forecasts, so an additional step is required to convert probabilistic forecasts into categorical forecasts to quantify their value.

1. Introduce a critical probability threshold $p_t$ to convert the probabilistic forecast into a deterministic forecast using the quantile function,

2. Construct a categorical forecast and contingency table from this deterministic forecast and apply Eq. (2) over a range of $\alpha$ as before,

3. Repeat step 1 and 2 for many probability thresholds over the range $0 \leq p_t \leq 1$ to form a set of possible REV values for each value of $\alpha$,

4. Take the maximum value from this set for each value of $\alpha$ to construct a single curve which envelopes the many curves from each value of $p_t$,

5. This envelope is then considered to represent the value of the forecast system.

Constructing an envelope to represent the forecast value of the system in step 4 can lead to a problematic interpretation. It implicitly assumes that the user will always self-calibrate to select the best critical threshold $p_t$ for their decision before the event has occurred. This is impractical and the method therefore leads to an over estimation of the expected forecast value. This envelope could be alternatively interpreted as the maximum attainable forecast value. The impracticality of this method is well understood (Zhu et al., 2002) but frequently ignored when applied in practice.

Step 1 of the approach models how decision-makers commonly make decisions using probabilistic forecasts. That is, before the event has occurred (ex ante) a decision-maker will choose a probability threshold that represents the degree of certainty they require to act. If the forecast probability of the event occurring is larger than this threshold then they will act. We refer to this as the *threshold-approach*.

Alternatively, one could set the critical probability threshold equal to $\alpha$ which assumes that decision-makers will self-calibrate based on an awareness of their specific exposure to damages (Richardson, 2000). When forecasts are perfectly reliable this approach is equivalent to the maximum forecast value from step 4 (Murphy, 1977). Forecast systems are not perfectly reliable however, even with contemporary post-processing methods (Li et al., 2016b; Woldemeskel et al., 2018; McInerney et al., 2020). The realised value curve will therefore lie below the maximum value curve when applied to real-world forecasts. This alternative approach does not appear to be commonly used by the water resource community to make decisions, or the verification community to assess them.

### 2.2 Expected Utility Theory approach

Matte et al. (2017) introduced a method to quantify forecast value based on expected utility maximisation with a state dependent utility. The method is flexible enough to model binary, multi-categorical, and continuous-value decisions, along





with risk averse decision-makers. The method assumes that decisions of how much to spend on mitigating damages are based on the forecast probability that the event will occur. We will refer to this approach to decision-making as the *optimisation-approach* to contrast it with the *threshold-approach*.

For a general decision problem with multiple possible future states of world, the following equation specifies the von Neumann-Morgenstern expected utility for a single timestep over $M$ states.

$$U\left(\tilde{E}_t\right) = \sum_{m=1}^{M} p_t^m \mu\left(E_t^m\right) \qquad (4)$$

where $p_t^m$ is the probability of state $m$ occurring in timestep $t$ and $E_t^m$ is the outcome associated with that state. The outcome is typically but not necessarily in monetary units. A utility function $\mu(\cdot)$ maps the outcome to a utility. This utility represents

an ordinal value that the decision-maker gains from that outcome occurring. The expected utility $U\left(\tilde{E}_t\right)$ can be considered a probability weighting of the transformed outcomes of all possible states of the world.

Risk aversion is represented by the concavity of $\mu(\cdot)$, such that when a decision-maker is risk averse the utility gained from an extra dollar is less than the utility lost when losing a dollar (Mas-Colell, 1995). Therefore, on average the risk is only worth taking when the probability of gaining an extra dollar is more likely than losing a dollar; this is known as the

probability premium. Absolute risk aversion is suitable for the comparison of options whose outcomes are absolute changes in wealth, and relative risk aversion where outcomes are percentage changes in wealth. The degree of aversion could be constant, increasing, or decreasing with respect to wealth. A consumer or investor generally takes more risks as they became wealthier, and their preferences can be reasonably approximated by *decreasing absolute risk aversion*.

Matte et al. (2017) assumes that on average a public agency water manager is more likely to exhibit *constant absolute risk*

*aversion* (CARA). For example, we assume that their preference for precise forecasts (risk aversion) remains fixed even if the possible losses from one decision are much larger than another. In this case a utility function satisfying these properties can be defined by

$$\mu(E) = -\frac{1}{A}\exp(-A \cdot E) \qquad (5)$$

where $A$ is the Arrow-Pratt coefficient of absolute risk aversion and $E$ is the economic outcome (Mas-Colell, 1995).

Babcock et al. (1993) cautions against interpreting the risk aversion coefficient directly and notes the importance of considering how perception of risk aversion depends on the possible loss. A more interpretable measure which allows comparison between studies with different losses is the *risk premium*; the proportion of loss a decision-maker would pay to eliminate a decision and replace it with a certain outcome. The method introduced here can use any utility function, such as *constant relative risk aversion* which was used by Katz and Lazo (2011).

The economic model used in this study is a simplified version of that used by Matte et al. (2017) which determines the net outcome from a cost-loss decision to allow systematic comparison with REV, however any economic model can be used.





The model used by Matte et al. (2017) can consider intangible damages, distributing spending over multiple lead-times, and calibration to monetary units, and damages informed by flood studies. We are primarily interested in a relative measure of forecast value which can be used more generally for different decision-makers and locations rather than the absolute monetary value of a specific decisions.

For a state of the world $m$ at a specific timestep $t$, with damages $d(m)$, cost to mitigate the damages $C_t$, and amount of damages avoided $b_t(m)$, the outcome is given by

$$E_{t,m} = b_t(m) - d(m) - C_t \tag{6}$$

The benefit function $b(m)$ specifies the damages avoided from taking action to mitigate them,

$$b_t(m) = \min(\beta \cdot C_t, d(m)) \tag{7}$$

where the spending leverage parameter $\beta$ controls the extra damages avoided for each dollar spent. This is a similar concept (albeit inverted) to the cost-loss ratio $\alpha$ in the REV metric. The damage function $d(m)$ relates the streamflow magnitude to the economic damages and must be specified for the decision of interest. This economic model assumes that benefits increase linearly as more is spent on damage mitigation, followed by a loss if the spend amount is greater than the damages. The optimal amount $C_t^\pi$ to spend at timestep $t$ can be found by maximising the expected utility following substitution of Eq. (5)-(7) into Eq. (4),

$$
\begin{aligned}
C_t^\pi &= \underset{C_t}{\operatorname{argmax}} \ U(\tilde{E}_t) \\
&= \underset{C_t}{\operatorname{argmax}} \ \sum_{m=1}^{M} -\frac{p_t^m}{A} \exp\left[ -A \cdot \left( \min(\beta \cdot C_t, d(m)) - d(m) - C_t \right) \right]
\end{aligned} \tag{8}
$$

This optimal spend amount for each timestep must be found ex ante, that is before the event has taken place, when the future state of the world is unknown, but a forecast is available. The probabilistic forecast (for some lead-time) is used to determine the forecast likelihood of each state occurring and calculate the ex ante expected utility $U(\tilde{E}_t)$ in Eq. (8). The optimal amount to spend on mitigation is the amount which leads to the largest ex ante expected utility.

The utility can also be calculated ex post, after the event has taken place, and a singular state of the world is known (streamflow observation). This leads to the following expression for the ex post utility after substitutions into Eq. (4)

$$\Upsilon(E_t) = \mu\left( \min(\beta \cdot C_t^\pi, d(m_t^o)) - d(m_t^o) - C_t^\pi \right) \tag{9}$$

where $\Upsilon(E_t)$ is the ex post utility, $C_t^\pi$ is the spend amount that was found ex ante, $m_t^o$ is the state of the world associated with the observed flow at timestep $t$. The ex post utility quantifies the benefit a decision-maker would have gained if they



spent $C_t^\pi$ on mitigating the damages which occurred as a result of the observed flow. It's important to note that since utility is an ordinal quantity that represents a decision-maker's preference over the possible decision outcomes, the utilities can be

compared but the actual value is noninterpretable. The ex post utility is used in the RUV metric introduced in Sect. 3 .

Three ex post metrics were used in Matte et al. (2017) to quantify forecast value using spend amounts found ex ante. They use economic variables (utility, avoided losses and amount spent) averaged over forecasts spanning an historical period. None of these metrics are equivalent or directly comparable to REV and results were not parameterised by an equivalent of the cost-loss ratio. The mathematical form and interpretation of these 3 metrics are included in the Supplement.

Expected Utility Theory can be used to model more decisions with more realism than is possible with the strong assumptions of REV. However, the economically relevant metrics and parameterisation used to quantify forecast value by Matte et al. (2017) pose a challenge when comparing the outcomes from the two methods.

## 3        Relative Utility Value

This section introduces a new metric which allows direct comparison of the results quantified by the two alternative forecast

value approaches described in Sect. 2 . It aligns the two approaches and allows comparison using the Value Diagram, which is familiar to the environmental modelling verification community and a compelling communication tool. RUV is inspired by REV and skill scores, but with terms based on the ex post expected utility.

$$\text{RUV} = \frac{\mathbb{E}_{t\in T}\left[\Upsilon\left(E_t^r\right)\right] - \mathbb{E}_{t\in T}\left[\Upsilon\left(E_t^f\right)\right]}{\mathbb{E}_{t\in T}\left[\Upsilon\left(E_t^r\right)\right] - \mathbb{E}_{t\in T}\left[\Upsilon\left(E_t^p\right)\right]} \tag{10}$$

where $\mathbb{E}_{t\in T}\left[\Upsilon\left(E_t^\bullet\right)\right]$ is the expected value of the ex post expected utility from Eq. (9) over a set of observations and either

forecast ( $f$ ), reference climatology ( $r$ ), or perfect information ( $p$ ). A nice feature of RUV is that it uses the whole probabilistic forecast and does not first convert it to a deterministic forecast like REV.

RUV has all the benefits and familiarity of REV but is a more flexible way to quantify forecast value. Any economic model or form of risk aversion can be used to construct the expected utility terms required by RUV because it is built on the Expected Utility Theory framework. In this paper we focus on the method with the economic model detailed by Eq. (6)and

(7), and risk aversion in Eq. (5). If RUV is parameterised using $\beta = \dfrac{1}{\alpha}$ and visualised on a Value Diagram it can be interpreted in the same way as an REV curve. The flexibility of the utility framework allows the user to make explicit choices about suitable approximations to model the decision problem. This can be accomplished by modifying the economic model, damage function and risk aversion through Eq. (5), (6) and (7) when used to calculate RUV. These assumptions can then be evaluated and extended with additional information if available. Unlike REV using Eq. (2), additional evaluation

information is available for each timestep such as the amount spent, damage avoided and economic utility.





### 3.1 Relationship between RUV and REV

Figure 2 contrasts the processes used by REV and RUV to quantify the value of probabilistic forecasts. Note that RUV uses the same inputs as REV and leads to the same output, however RUV allows the economic model, damage function and risk aversion to be explicitly specified. The internal process is very similar except RUV maximises utility rather than minimises

expense.

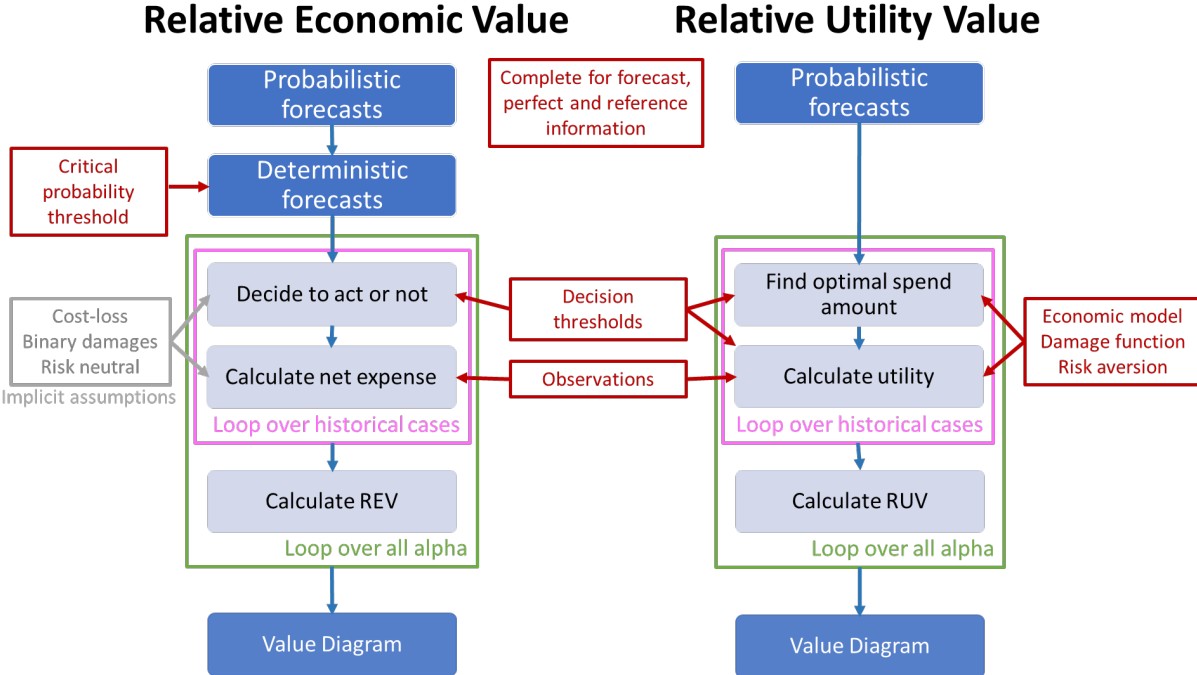

**Figure 2: Flowcharts showing the process followed to quantify the value of probabilistic forecasts using either RUV with an optimisation approach to decision making, or REV using the threshold-approach with a specific critical probability threshold. The**
**sub-processes in the pink boxes are repeated for forecast, perfect, and reference information before being used to calculate REV and RUV. In practice, Eq. (2) is used to calculate REV which is based on a contingency table with an assumption that it has converged to the long-run performance of the system.**

Unlike REV, there is no analytical solution for RUV due to the optimisation step in Eq. (8) unless assumptions are placed on

the decision context. When the following 5 assumptions are applied to RUV it is equivalent to REV.

      1.   Binary damage function is used which is a positive value for the losses above the decision threshold, and 0 otherwise,

      2.   Decision-makers are risk neutral as specified by a linear utility function,

      3.   Forecasts are deterministic with the probability of flow above the threshold always either 1 or 0,

4.   The historical frequency of the binary event is used as the reference baseline,



5. All possible losses are avoided.

The mathematical justification for these assumptions and a proof of the equivalence is detailed in Appendix A and the Supplement. Note that when applying these assumptions, the core RUV method illustrated in Figure 2 remains the same but the probabilistic forecast is first converted to a deterministic forecast. Table 2 summarises how decision concepts are represented in each forecast value method and demonstrates the enhanced flexibility of the RUV metric.

**Table 2: Comparison of REV and RUV Forecast value methods for defining decisions and decision-maker characteristics**

|  | **Relative Economic Value (REV)** | **Relative Utility Value (RUV)** |
|---|---|---|
| **Level of damages** | Fixed loss (dimensionless) Equivalent to step damage function | Damage function is flexible and can be tailored to decision |
| **Mitigation of damages** | Fixed cost (dimensionless) Equivalent to fixed spend amount | Spend amount is optimised and varies at each timestep |
| **Exposure to damages** | Cost-loss ratio | Spending-leverage parameter |
| **Aversion to risk** | Always risk neutral | Level of risk aversion and type of utility function can vary |
| **Decision types** | Binary | Binary, multi-categorical or continuous-value |
| **Forecast value baseline** | Historical event frequency | Any alternative forecast |
| **Probabilistic decision-making** | Threshold-approach | Optimisation-approach or threshold-approach |
| **Economic model** | Fixed cost-loss | Economic model is flexible and can be tailored to decision |
| **Interpretation** | Value Diagram | Value Diagram |

## 4      Methodology

A case study is used to determine how the value of probabilistic forecasts change with different decision types, decision-makers, and decision-making approaches. A targeted approach is adopted to contrast the RUV and REV methods and





illustrate the impact of decision characteristics, rather than an exhaustive evaluation of the value of the specific forecasts used.

## 4.1    Background

Water resource management and the equitable distribution of water to competing stakeholders is challenging due to long-term decreasing trends in available surface water (Zhang et al., 2016), increasing high intensity storm events (Tabari, 2020), river basins overallocated to irrigated agriculture (Grafton and Wheeler, 2018), and deteriorated river system dependant ecosystems (Cantonati et al., 2020). Such challenges to decision-making may be assisted by subseasonal streamflow forecasts and quantifying forecast value would help adoption. For forecasts to be useful they need lead-times long enough to

account for river travel-times and decision overheads. Subseasonal forecasts with lead-times out to 30 days would assist management of storages where a release leads to an impact far downstream. For example, agencies operating in the southern Murray-Darling Basin of Australia, such as the Murray-Darling Basin Authority (MDBA) and Goulburn-Murray Water (GMW), make such decisions and may benefit from streamflow forecasts for the Enhanced Environmental Water Delivery method (Murray–Darling Basin Authority, 2017). When operational decisions are informed with probabilistic forecasts the

threshold-approach is used with a set of fixed critical probability thresholds, and a degree of risk aversion is implicitly assumed (personal correspondence with MDBA). As far as the authors are aware, the relative value of streamflow forecasts for these decisions and decision-maker characteristics has not been previously quantified.

## 4.2    Location

Results are presented for the water level station Biggara (401012) on the Murray River in the southern Murray-Darling

Basin, Australia. Biggara is upstream of Hume Dam, a major reservoir used for environmental water releases, irrigated agriculture, and town supply. It is in a temperate region, has a contributing area of 1,257 km$^2$, a mean rainfall of 1,158 mm/year, mean runoff of 361 mm/year.

## 4.3    Streamflow forecasts

Daily streamflow forecasts are generated using the following method, which demonstrated good performance at subseasonal

time horizons in earlier studies (McInerney et al., 2020). We generated 30-day ensemble forecast time series (100 members) starting on the 1st of each month over the period 1991 to 2012. Raw streamflow forecasts were simulated using the GR4J rainfall-runoff model forced by rainfall from the Australian Community Climate and Earth-System Simulator Seasonal (ACCESS-S1) which had been post-processed using the Rainfall Post-Processing for Seasonal forecasts method (RPP-S) (Perrin et al., 2003; Hudson et. al., 2017; Schepen et al., 2018). Final streamflow forecasts were generated by post-

processing the raw forecasts using the Multi-Temporal Hydrological Residual Error (MuTHRE) model (McInerney et al., 2020). Post-processing ensured that the statistical properties of the forecasts closely match the observations for all lead-times and accumulations, leading to forecasts which are sharp, reliable, and unbiased. Forecasts with these characteristics can be





described as seamless in the sense that they perform well at different time horizons and time resolutions. Further information on these forecasts can be found in McInerney et al. (2020).

### 345    4.4    Decision types

A binary decision of flow exceeding a single threshold can be considered the simplest for a decision-maker to manage, flow exceeding the height of a levee for example. A multi-categorical decision with more than two classes introduces additional complexity for the decision-maker to consider. An example of this is a minor, moderate, and major flood classification which correspond to increasing categories of impact. A mitigation decision based on continuous-flow is the limiting case of
a very large number of flow classes. An example is adjusting dam releases to match storage inflow during flood operations. A binary decision has traditionally been used as the prototypical model of decision making in decision-theoretic literature and is a limiting assumption of REV (Katz and Murphy, 1997). Decisions involving more flow classes are more complex for decision-makers to reason about as there are more possible outcomes to consider, but they are an essential feature of many real-world decisions and cannot be ignored.

Three types of decisions have been included in the case study: (i) binary decisions with flow above a single threshold, either the top 25% of top 10% of the observation record; (ii) multi-categorical decisions with flow in 5 classes over a range of thresholds; and (iii) continuous-flow decisions using flow from whole flow regime. These thresholds are indicative of decisions which depend on moderate to high flow at Biggara, such as operational airspace management of the Hume Dam or minor inundation upstream of Yarrawonga Weir when coinciding with a dam release.

### 360    4.5    Economic damages

The relationship between damages and flow in Eq. (6) and (7) when applying the RUV metric is specified using a non-dimensional logistic function,

$$d(q) = \frac{\delta}{1 + \exp(-k(q - \tau))} \tag{11}$$

The logistic function can be parameterised to have very similar behaviour to the Gompertz curve used in flood damage
studies and used by Matte et al. (2017), with $d(q)$ representing the cumulative damages incurred from all flow up to $q$ (Li et al., 2016a). It was parameterised to reasonably characterise losses from high flow events; no damages when flow is zero, increasing quickly from around the top 20% of flow, and approaching 1 at very high values above the top 1% of flow. The following parameter set was found to be suitable; $\delta = 1$, $k = 1$ and $\tau$ equal to the value corresponding to the top 1% of observed historical flow, see Figure 3 and Sect. 6.3.



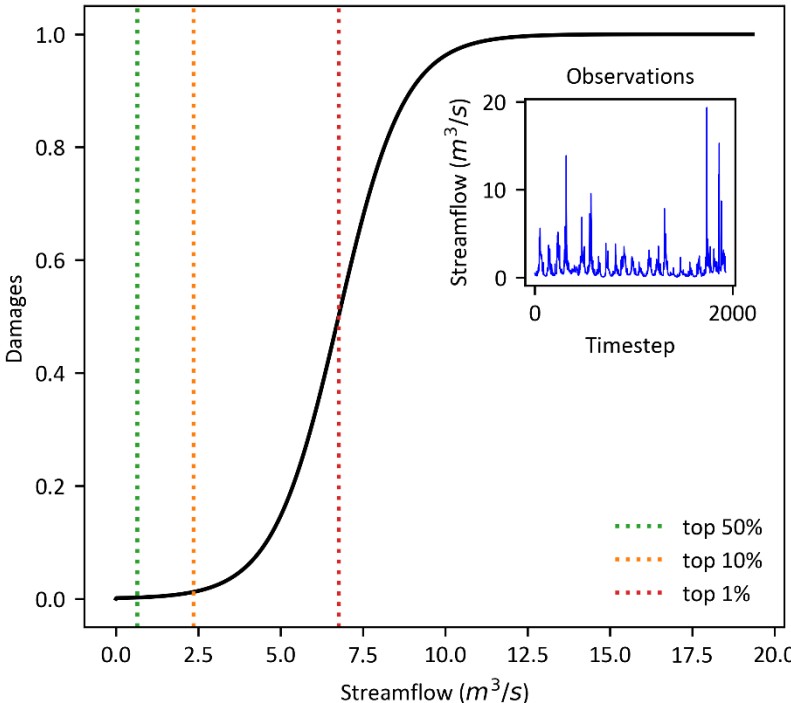


**Figure 3: Damage function used in the case study based on a logistic curve with an inflection point at the top 1% of observed flow.**

### 4.6  Risk aversion

It is difficult to precisely know a decision-maker's level of risk without a history of prior decisions. Moreover, it would be incorrect to assume that all decision-maker's share the same level of risk. Therefore, a range of risk aversions have been
considered to illustrate its impact on forecast value. In this study we have used risk aversion coefficients $A \in \{0, 0.3, 1, 5\}$ which correspond to risk premiums of $\theta \approx \{0\%, 15\%, 44\%, 86\%\}$ for a CARA utility function with maximum losses of $\delta = 1$ (Babcock et al., 1993). These represent decision-makers who are neutral, slightly, moderately, and highly risk averse. When risk premiums are considered, our range of risk aversion coefficients is similar to those used by Tena and Gómez (2008) and Matte et al. (2017). Finding appropriate values of risk aversion for a specific decision-maker is beyond the scope of this
study but would be highly beneficial in user-focused forecast value studies.

### 4.7  Experiments

The value of the subseasonal forecasts are quantified using the RUV and REV metrics. Experiments are performed over the dimensions of forecast lead-time, decision type, decision making approach, metric, and decision-maker risk aversion.
Streamflow forecasts from multiple daily lead-times were grouped together to quantify forecast value over 7-day and 14-day





forecast horizons. A fixed climatology based on all observed values in the record is used for the reference baseline of RUV to align with that used in REV. Table 3 summarises the specific attributes used for each figure, with the key dimension highlighted as red text.

**Table 3: Dimensions of forecast value problem used for each figure. Key dimension introduced in each figure is highlighted with red text.**

| Experiment purpose | Lead-times (days) | Decision type | Decision thresholds | Decision-making approach | Metric | Risk aversion |
|---|---|---|---|---|---|---|
| Experiment 1: Equivalence of REV and RUV, and impact of fixed probability thresholds. Moderate flow example. (Figure 4) | 1-7 | Binary | Top 25% | Threshold | REV RUV | 0 |
| Experiment 2: Contrast decision-making approaches. Moderate flow example. (Figure 5) | 1-7 | Binary | Top 25% | Threshold Optimisation | RUV | 0 |
| Experiment 3: Subseasonal forecast value for different decision types. High flow examples. (Figure 6) | 1-7 8-14 15-30 | Binary Multi-categorical Continuous-flow | Top 10% Top 20%, 15%, 10%, 5% All flow | Optimisation | RUV | 0 |
| Experiment 4: Impact of risk aversion of on forecast value. High flow examples. (Figure 7) | 1-7 | Binary Multi-categorical Continuous-flow | Top 10% Top 15%, 10%, 5%, 1% All flow | Optimisation | RUV | 0, 0.3, 1, 5 |
| Experiment 5: Key driver of impact of risk aversion on forecast value. (Figure 8) | 1-7 | Binary | All flow | Optimisation | RUV | 0, 0.3, 1, 5 |

## 5 Results

### 5.1 Experiment 1: Equivalence of RUV and REV, and impact of fixed probability threshold

In experiment 1, forecast value has been quantified using REV and RUV with the assumptions detailed in Sect. 3.1: binary damage function, risk neutral decision-maker, deterministic forecasts, event frequency for reference baseline, and all losses avoided. As expected, Figure 4 demonstrates that the results are identical between the two methods.





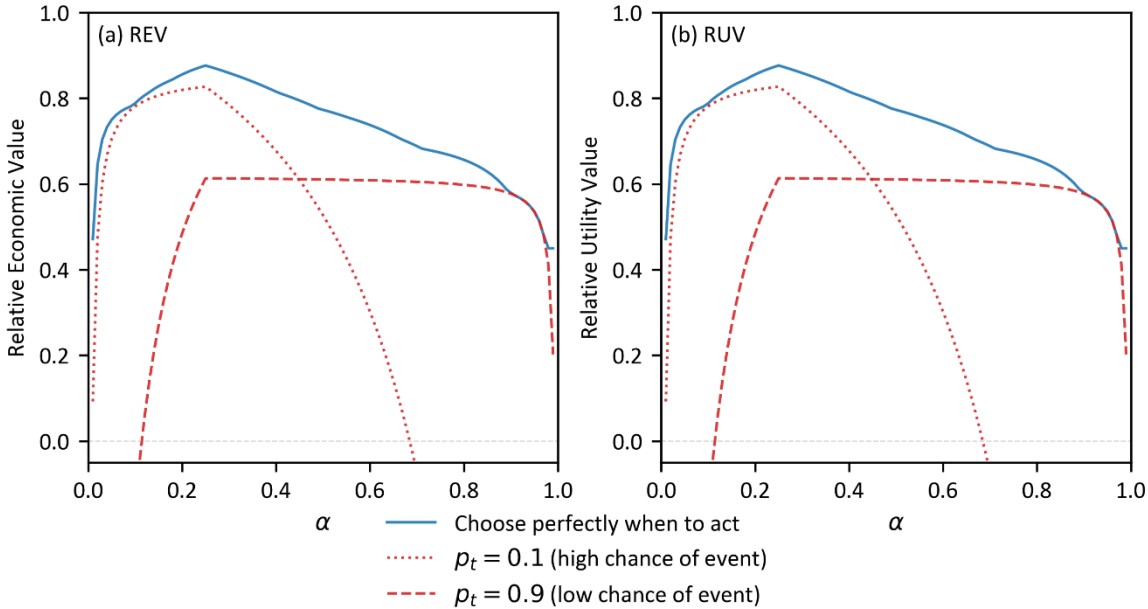

**Figure 4: Forecast value quantified using (a) REV and (b) RUV with assumptions enforced, and the threshold approach for decision-making. With a binary decision of flow exceeding the top 25% of observations, subseasonal forecasts from the first week of lead-times, and a risk neutral decision-maker. Critical probability thresholds for the three curves are the value leading to maximum forecast value, and the 0.1 and 0.9 forecast quantile, corresponding to acting when there is a high or low chance of event occurring respectively.**

We now explore the detrimental impact on forecast value of using the threshold-approach to convert probabilistic forecasts to deterministic forecasts. Any forecast value method using the threshold-approach needs to select a critical probability threshold $p_t$ to convert probabilistic forecasts to deterministic forecasts. Figure 4 includes three curves corresponding to decisions made with different thresholds. The blue line shows the value obtained when the threshold $p_t$ is chosen to maximise that value at each $\alpha$ (see Sect. 2.1.2). This is an upper limit that cannot be obtained in practical situations because it implies a decision-maker has either perfect foresight or a perfectly reliable forecast, and $P_t = \alpha$ will lead to maximum value if the forecast is perfectly reliable (Richardson, 2000). The red lines show how the choice of $p_t$ can have a dramatic impact on the value of forecasts for a decision, with the dotted line showing forecast value when $p_t = 0.1$ and dashed line when $p_t = 0.9$. RUV is negative for some regions of $\alpha$ which indicates that those decision-makers should use the climatological baseline rather than the forecasts when making decisions.

This result clearly shows that to extract the most value from forecast information a decision-maker needs to consider their exposure to damages $\alpha$ when choosing $p_t$. For example, when a decision-maker with $\alpha = 0.8$ uses $p_t = 0.9$ they gain significant value from the forecasts ($RUV \approx 0.6$), but if they use $p_t = 0.1$ their outcome using forecasts is worse than using the reference climatology ($RUV < 0$), while for a different decision-maker with $\alpha = 0.1$ the opposite is true. It additionally





shows that the Value Diagram used with REV remains a compelling way to visualise how RUV forecast value varies for different decision-makers.

This result and the derivation in Appendix A demonstrate that RUV and REV are equivalent when appropriate assumptions are imposed. It demonstrates that REV can be considered a special case of the more general RUV metric.

### 5.2 Experiment 2: Contrasting the threshold-approach and optimisation-approach for decision making

Figure 5 adds two more forecast value curves, generated using RUV, to Figure 4. The black line shows value when the optimisation-approach is used to make spending decisions with the subseasonal forecasts (detailed in Sect. 2.2), and the grey

line shows value when the threshold-approach is used with $P_t = \alpha$. The result demonstrates that making decisions using either approach provides close to the maximum value possible for all decision-makers (different values of $\alpha$). This contrasts dramatically with the threshold-approach using specific fixed values for $p_t$ (red lines) which only provides maximum value for a very small range of decision-makers.

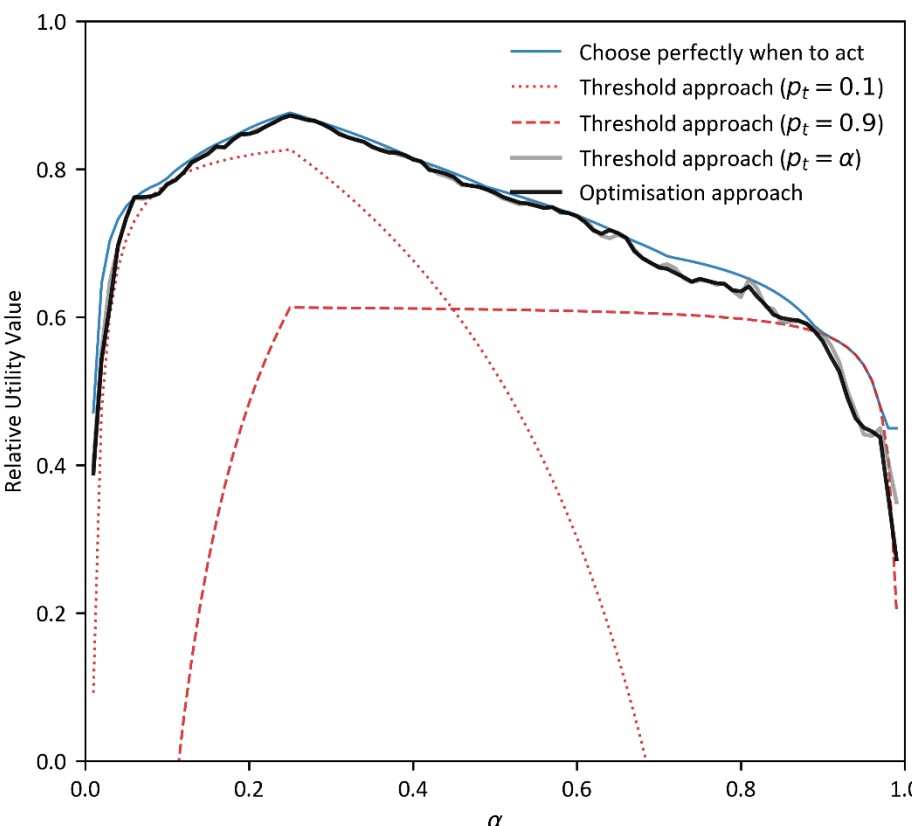

**Figure 5: Forecast value quantified using four different approaches to decision-making: the optimisation-approach and the threshold-approach with either perfect critical probability thresholds, specific critical thresholds, or the critical threshold set equal to the alpha value. A binary decision of flow exceeding the top 25% of observations was used, with subseasonal forecasts from the**




**first week of lead-times, and a risk neutral decision-maker. Specific critical thresholds are the 0.1 and 0.9 forecast quantile, corresponding to acting when there is a high or low chance of the event occurring respectively.**

Investigations (not shown) indicated that the optimisation and $P_t = \alpha$ curves (black and grey lines) are non-smooth because

of the limited number of events in the observation record, and the small difference between the grey and black lines is due to

ensemble sampling error. It is notable that forecast value from these two different decision-making approaches are

essentially equivalent as illustrated by the closeness of the black and grey lines in Figure 5. Additional analysis (not shown)

found this equivalence to be robust to the type of decisions (binary, multi-categorical, or continuous-flow) and changes in

forecast reliability.

### 5.3    Experiment 3: Comparing Forecast value for different types of decisions

Figure 6 presents results for binary, multi-categorical and continuous-flow decisions in separate panels. RUV was calculated

for the daily subseasonal forecasts with lead-times pooled from the 1st week (blue lines), 2nd week (orange lines) and 3rd and

4th weeks combined (green lines). The decision-maker is assumed risk neutral, and the optimisation-approach was used.

Overall, the forecasts provide excellent value for these three different decision types over all time-horizons (max 30 days),

implying that any decision-maker would likely benefit from using the forecast information over the climatology baseline.

Peak RUV was over 0.8 in the first week for all decision types, and close to 0.7, 0.6, and 0.5 in subsequent weeks for binary,

multi-categorical, and continuous-flow decision types respectively. The exception is for decision-makers with high exposure

to damages in the 3rd and 4th weeks, where RUV drops below zero above $\alpha = 0.6$ for binary decisions and $\alpha = 0.9$ for

multi-categorical decisions.

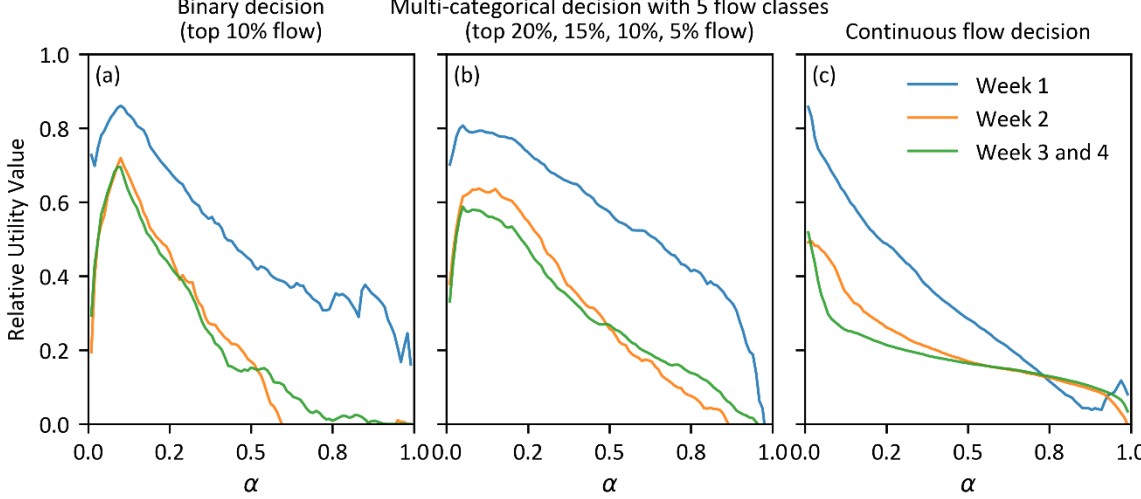

**Figure 6: Forecast value for (a) binary decision of flow exceeding the top 10% of observations, (b) flow within 5 classes with thresholds at the top 20%, 15%, 10% and 5% of observations, and (c) continuous-flow. Decisions are made using the optimisation-approach for decision-making with a risk neutral decision-maker, and subseasonal forecasts for the 1st, 2nd and combined 3rd and 4th weeks of lead-times.**





Almost all decision-makers will experience positive value from incorporating the streamflow forecasts into their decisions across all lead-times and decision types. The only decision-makers who should avoid using the forecasts in all cases are those with very large exposure to damages, a common finding in studies using REV (Roulin, 2007). However, there is important variation in RUV across $\alpha$, lead-time, and decision type. For example, beyond the second week decision-makers

with $\alpha > 0.6$ should prefer the reference climatology for the binary decision and prefer forecasts for the multi-categorical and continuous-flow decisions. Regardless of the decision type or lead-time, forecasts provide maximum value for decision-makers with $\alpha$ close to the probability of the most damaging flow class occurring. For example, for the binary decision the peak RUV value is located at $\alpha = 0.1$ which corresponds with the event frequency of decision threshold used (top 10% of flow). Forecast value at small $\alpha$ is enhanced for continuous-flow decisions relative to the other decision-types. This seems

to be because large damages from infrequent extreme events are more adequately mitigated in continuous-flow decisions because a correspondingly large amount is spent when they are forecast correctly.

## 5.4 Experiment 4: Impact of risk aversion

Experiment 4 contrasts forecast value for a risk neutral decision-maker against 3 different levels of risk aversion for binary, multi-categorical, and continuous-flow decisions. The results presented in Figure 7 for the RUV metric (first row) as well the

overspend (middle row) and utility-difference metrics (last row) used by Matte et al. (2017) which provide insight into the spending decisions and utility respectively. By varying $A$ in Eq. (5) risk aversion is found to have a moderate impact on the value of forecasts for the multi-categorical and continuous-flow decisions, and a minor impact for binary decision types (see Figure 7 first row). Increased risk aversion shifts the RUV curve toward users with higher $\alpha$, suggesting that risk averse decision-makers with more exposure to damages would benefit more from using forecasts to make their decisions.



**Figure 7: RUV, overspend and utility-difference for different levels of decision-maker risk aversion, for a binary decision of flow exceeding the top 10% of observations (first column), flow within 5 classes with thresholds at the top 15%, 10%, 5%, and 1% of observations (middle column), and continuous-flow (last column). Decisions made using the optimisation approach with subseasonal forecasts from the 1st week of lead-times.**

The overspend (middle row) and utility-difference results (last row) indicate that risk aversion has a minor impact on the spending decisions and the resultant utility. The overspend panels show that regardless of risk aversion, on average a decision-maker will spend more than necessary when their cost of mitigation is small relative to the potential avoided losses





(small $\alpha$ ). Conversely, when $\alpha$ is large they will underspend on average. When risk aversion is increased, decision-makers spend increasingly more.

**5.5      Experiment 5: Mechanism behind the varying impact of risk aversion**

It is notable that the impact of risk aversion in Figure 7 is different for each decision type; minor for the binary decisions, moderate for multi-categorical and continuous-flow, and particularly enhanced for highly risk averse decision-makers. Experiment 5 investigates the mechanism behind this. Figure 8 presents the difference in RUV between risk averse and risk neutral decision-makers (y-axis), for a binary decision at a single exposure to damages of $\alpha = 0.2$ . The binary decision

threshold (x-axis) is varied from the $0.1 - 16$ m³/s (bottom 2% to top 0.04%) and decisions are made using the optimisation approach with subseasonal forecasts from the 1ˢᵗ week of lead-times. This contrasts with the binary decision in experiment 4 where the decision threshold is fixed at 2.4 m³/s (top 10%) and $\alpha$ is varied.

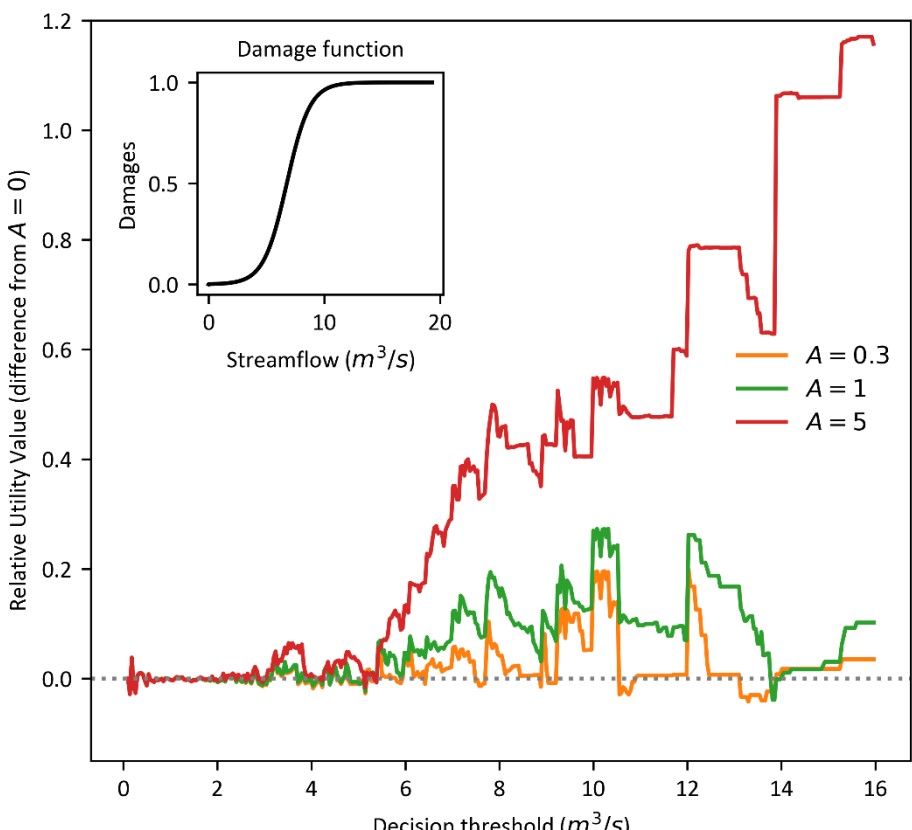

**Figure 8: difference in RUV between risk averse ( $A > 0$ ) and risk neutral ( $A = 0$ ) decision-makers (y-axis), for a binary decision**
**at a single exposure to damages of $\alpha = 0.2$ . The binary decision threshold (x-axis) is varied from 0.1 − 16 m3/s and decisions are made using the optimisation approach with subseasonal forecasts from the 1st week of lead-times.**



Below a critical decision threshold of approximately 5 m³/s (top 2% flow) the difference in RUV between any level of risk aversion and risk-neutrality is negligible. Above this value an increasing difference is clear, particularly in the highly risk

averse case, with risk averse decision-makers gaining more value from the forecast information than risk neutral. This finding was consistent for multi-categorical decisions of any number of flow classes and all values of $\alpha$ except at extreme high and low values (not shown). It demonstrates that the decision thresholds used, specifically in relation to the damage function, are the key drivers behind the impact of risk aversion regardless of the decision type. The difference in impact of risk aversion across the different decision types in Figure 7 can therefore be explained by the specific decision thresholds

used in relation to this critical value; the binary decision threshold of 2.4 m³/s used in experiment 4 was less than the critical value of 5 m³/s and only a minor impact from risk aversion was found, whereas the top decision threshold for the multi-categorical decision was 6.8 m³/s, above this critical value, and a moderate impact was found, and an even larger impact was found for the continuous-flow decision which includes contribution from the largest flows.

## 6    Discussion

According to statistical forecast verification metrics, probabilistic streamflow forecasts have been shown to be skilful and statistically reliable (McInerney et al., 2021; Li et al., 2016b). However, their ability to improve decision outcomes has not been extensively established. Additionally, REV, the most frequently used forecast value method, can only be applied to a limited number of real-world decisions. In this paper we develop a new forecast value method, Relative Utility Value (RUV), which is more flexible than REV and can applied to more decision. The flexibility of RUV is demonstrated with a

case study using probabilistic subseasonal streamflow forecasts to inform binary, multi-categorical, and continuous-flow decisions with risk averse decision-makers. The 5 experiments reported in Sect. 5  systematically explore the impact of different aspects of a decision on forecast value: the forecast value method, the probabilistic decision-making approach, types of decisions, decision-maker risk aversion, and the mechanism behind varied risk aversion impact. First, we find that under certain conditions RUV and REV are equivalent, and REV can be considered a special case of the more general RUV

method (see Figure 4 and Appendix A ). Second, making decisions with fixed critical probability thresholds is leads to maximum forecast value only for a very small set of users, and using an optimisation-based approach makes better use of probabilistic forecast information (see Figure 5). Third, subseasonal forecasts offer more value than a climatological average for almost all lead-times and decision-makers regardless of the decision-type (see Figure 6). And finally, risk aversion has a minor to moderate impact on forecast value (see Figure 7) but the degree of impact is sensitive to the decision context being

evaluated. The key mechanism driving this impact is the decision thresholds used relative to the damage function (see Figure 8). This section interprets these results through the lens of forecast users and producers.





## 6.1    Benefits of RUV over alternatives

1. *Forecast value complements forecast verification.* Unlike forecast verification, forecast value considers the broader context within which decisions are made. This allows forecast producers, such as the Bureau of Meteorology, to understand their customer impact by evaluating service enhancements against user decisions. Determining which method or enhancement to operationalise is typically made using forecast verification as a key deciding factor. Quantifying the value of forecasts based on impact offers a complementary line of evidence which places the forecast user at the centre of the conversation. Because RUV encourages a dialog between the forecast producer and user to define the full decision context it may enhance communication and service adoption. For forecast users, it provides a new capability: an evidence-based approach to decide which forecast information and decision-making process will improve their outcomes. For example, the Biggara case study in Sect. 5  indicates that subseasonal forecasts offer better value than climatology and that an optimisation-approach is beneficial when deciding to take early action to mitigate damages from a high flow event a few weeks ahead (see Figure 5 and Figure 6).

2. *RUV is more flexible than REV.* It can model more decisions with sufficient realism than REV because it explicitly specifies decision type, risk aversion, economic model, and decision-making approach. Real-world decisions may be binary, multi-categorical, or based on continuous-flow, and using a binary model (as in REV) in all cases will provide a misleading measure of forecast value for non-binary decisions. Figure 6 shows that neglecting this would have important implications for decision-makers; forecasts beyond week 2 should be used for the multi-categorical and continuous-flow but not for the binary decision (when $\alpha > 0.6$). Similarly, neglecting the realism of other aspects of the decision may lead to other misleading conclusions. The flexibility of RUV allows the user to decide how much realism to include in the forecast value assessment depending on the information available and tailor it to the decision context.

3. *RUV evaluates forecast value conditioned upon decision-makers exposure to damages.* Unlike single-valued metrics, common in traditional forecast verification, RUV is evaluated for wide range of decision-makers' exposure to damages, as is shown in the value diagram (Figure 1). This offers valuable insight that would otherwise be hidden. In particular, it is useful for forecast producers who can quickly compare one forecast system to another for a range of different users with different exposures to damages. However, this does make it comparatively more difficult to summarise and aggregate. To assist interpretation for a single forecast user, it is important they narrow the range of $\alpha$  which is relevant to their decision by considering their specific exposure to damages.

## 6.2    Implications of case study results

1. *Optimisation based decision-making is better than fixed critical probability thresholds when using probabilistic forecasts.* Figure 5 demonstrates that a specific critical probability threshold will only be optimal for a specific exposure to damages ($\alpha$ ) and suboptimal for all other values. When a decision-maker is choosing between using





the forecast or the climatological reference, they may choose incorrectly if their critical probability threshold is not aligned with their exposure to damages. This incorrect choice will be due to a deficiency in the threshold-approach to decision-making rather than the forecast information. This RUV based finding is well supported by the REV literature (Richardson, 2000; Wilks, 2001; Zhu et al., 2002; Roulin, 2007). A perfect critical probability threshold is typically used with REV (Figure 4), unfortunately this is not possible to achieve in practice and the quantified value is unrealistically high. Matte et al. (2017) introduced an optimisation-approach and we extended it here to further evaluate the impact on forecast value. This flexible approach makes best use of the forecast information available and is equivalent to the threshold-approach when the threshold is set equal to the decision-makers' exposure to damages $\alpha$ (Figure 5). When forecasts are reliable this method yields value which is very close to the maximum possible and forecast users may consider adopting this alternative approach for daily operational decisions. A Decision Support System would be required so the optimal amount to spend on mitigation can be calculated each time a new forecast is issued.

2. *Forecast information is more valuable for risk averse users making high-stakes decisions.* Figure 7 demonstrates that for a given forecast, a more risk averse decision-maker spends more to mitigate a potential damaging event than a less risk averse decision maker, all else being equal. This behaviour is consistent with their risk aversion because it leads to a more certain result, with the net outcome equal to the spend amount whether the event occurs or not. There is a large difference in impact of risk aversion for the different decision types however and Figure 8 summaries the findings of an investigation into this. Decision thresholds corresponding to very high flows lead to a larger impact. This finding explains why risk aversion has a large impact for the continuous-flow decision, spanning the whole regime, and a negligible impact for the binary decision with a single moderately high decision threshold. It may also explain apparently contradictory findings on the impact of risk aversion in the literature. Matte et al. (2017) assessed the impact of risk aversion on a multi-categorical decision (using overspend and utility metrics) and found it had a moderate impact (similar to the multi-categorical decisions shown in Figure 7e and Figure 7h). Their study used 12 uniformly spaced flow divisions over a high flow range and a damage function based on empirical flood studies, whereas this study used 4 widely spaced thresholds over a similar high flow range. A recent study by Lala et al. (2021) found minor impacts from risk aversion for binary cost-loss decisions with extreme rainfall forecasts using the same expected utility maximisation framework from Matte et al. (2017) and found a similar impact to Figure 7a. An alternative argument using reasoning from decision theory suggests that for a given risk premium the impact should be larger when decision thresholds are closer together (Mas-Colell, 1995). However, when investigated we found no evidence to support this. Further research to better characterise the response for different decision contexts would be useful because the impact is modulated by both the decision thresholds and the specific damage function, consideration of the inherent sampling error introduced for extreme events would also be useful.





### 6.3    Limitations and future work

Future work on the RUV metric will focus on the following aspects:

1.  *Exploring the impact of alternative damage functions, economic models, and utility functions on forecast value.*
    This manuscript focused on the impact of alternative decision types and risk aversion, and a comparative study of
    RUV and REV. The foundation in Expected Utility Theory allows us to model more decisions more realistically
    than REV, but it requires more information. When this information is unavailable or uncertain the user needs to
    apply assumptions, but it is not clear what the best strategy to take is. One strategy is to model all decisions as
    binary, cost-loss, and risk neutral and effectively convert RUV to REV. This study explores the implications of
    relaxing some, but not all, of those assumptions but is limited to a single case study. In particular, the damage
    function used was parameterised to simplify the introduction of RUV, facilitate comparison with REV, and
    highlight important implications for future studies. Further work will consider the impact of alternative damage
    functions and economic models tailored to other decision contexts. More descriptive economic models than cost-
    loss will be essential to consider decisions which involve non-economic intangible externalities like social, cultural,
    and ecological factors (Jackson and Moggridge, 2019; Expósito et al., 2020).

2.  *Expected Utility Theory approximates decision-making and contemporary frameworks may enhance the capability
    of RUV to model real-world decisions.* There is general agreement, and a substantial body of evidence, that
    Expected Utility Theory does not adequately describe individual choice (Kahneman and Tversky, 1979; Harless and
    Camerer, 1994). Many alternative models have been proposed which address these violations, such as Cumulative
    Prospect Theory (Tversky and Kahneman, 1992). Future work will consider whether quantifying forecast value
    using a foundation built on a better model of decision-making changes the conclusions reached.

3.  *Exploring the relationship between forecast value and forecast skill.* Roebber and Bosart (1996) found that
    statistical performance metrics were poor at predicting the cost-loss value of meteorological forecasts for several
    real-world decisions. The relationship was impacted by the decision-maker's $\alpha$ value, and when in aggregate, the
    distribution of $\alpha$ over all users. Using a real-time optimisation system to manage reservoir operations Peñuela et al.
    (2020) quantified forecast value through improvement in pumping costs and resource availability relative to a
    baseline. They found a relationship between forecast value and CRPS skill score mediated by user priorities and
    hydrological conditions. Although a relationship exists it is clearly mediated by the characteristics of the decision
    and decision-maker and in many cases forecast skill is not a good proxy for forecast value (Murphy and
    Ehrendorfer, 1987; Wilks and Hamill, 1995; Roebber and Bosart, 1996; Roulin, 2007; Peñuela et al., 2020).
    Exploring this relationship is of interest because the decision and decision-maker characteristics are made explicit in
    RUV. Converting RUV to a single-value metric by placing assumptions on the distribution of $\alpha$ could assist and
    additionally allow its use as an objective function for model calibration or as a summary statistic, Wilks (2001)
    considers this using REV.





## 7 Conclusions

Probabilistic forecasts have the potential to benefit water-sensitive decisions, such as operational water resource management and emergency warning services, but to date their value for decision making has not been established. Forecast value methods attempt to quantify this potential. However, the most commonly used existing method to evaluate forecast value, Relative Economic Value (REV), is only suitable for specific decisions. REV is unsuitable for many real-world decisions and when applied may lead to misleading conclusions on when to use forecasts. This manuscript introduces the RUV metric which has the same interpretation as the commonly used REV metric but is more flexible and can be applied to a far wider range of decisions. This is because many aspects of the decision-making process can be incorporated by the user and adjusted to match real-world decisions. These include the economic model, damage function, decision type, and decision-maker characteristics and preferences, such as risk aversion and exposure to damages. Importantly, we show that REV can be considered a special case of the more general RUV, when applying specific restrictive assumptions.

A case study demonstrates that subseasonal streamflow forecasts should be preferred over a reference climatology forecast for all lead-times studied (max 30 days) and almost all decision-makers regardless of their risk aversion. This positive forecast value is robust to changes in decision-maker characteristics, decision types (binary, multi-categorical, and continuous-flow), and decision-making approaches. However, beyond the second week, RUV indicates that decision-makers who are highly exposed to damages should use the reference climatology rather than the forecasts for the binary decision. This is not the case for the multi-categorical and continuous-flow decision however, where forecasts should be preferred. In this study, risk aversion is found to have a larger impact for multi-categorical and continuous-flow decisions than for binary decisions. However, this difference in impact is found to be a result of the specific decision thresholds used relative to the damage function rather than the decision type itself. With probabilistic forecasts, decisions are commonly made by first applying a fixed critical probability threshold. We find that this fixed threshold-approach to decision-making leads to sub-optimal use of the forecast information. Alternatively, an optimisation-approach which finds the ideal amount to spend on each decision leads to the best use of the forecasts. This difference suggests the importance of modelling the real-world decision-making approach when quantifying forecast value. RUV was used to model both decision-making approaches in this study.

RUV presents an opportunity to tailor forecasts and their assessment to the specific decisions, decision-making approach, characteristics, preferences, and economics of the decision-maker. It is hoped that this capability may encourage the assessment of forecast systems through the lens of customer benefit and be seen as a complement to forecast verification. This may lead to increased adoption of forecasts through deeper dialog and understanding, and ultimately to improved water resource management decisions.





## Appendix A    Proof of equivalence of REV and RUV under specific assumptions

This section demonstrates the equivalence of the REV metric as detailed in Eq. (2) and the RUV metric introduced in Sect. 3 when 5 assumptions are applied to the decision context. A complete derivation is included in the Supplement.

In a cost-loss decision problem the two relevant states are "flow above" and "flow below" a decision threshold $Q_d$.

$$
\begin{aligned}
m &= \text{above} && \text{if } Q_t \geq Q_d \\
m &= \text{below} && \text{if } Q_t < Q_d
\end{aligned}
\tag{12}
$$

**Assumption 1:** A step damage function with binary values of 0 and L is used to specify the losses above and below the decision threshold,

$$
d(m) = \begin{cases} L & \text{when } m = \text{above} \\ 0 & \text{when } m = \text{below} \end{cases}
\tag{13}
$$

To calculate the net outcome when action is taken to mitigate the loss, we substitute Eq. (7) and (13) into Eq. (6), which leads to the following net outcomes for the two states.

$$
\begin{aligned}
E_{t,above} &= \min\left(\beta \cdot C_t, L_t\right) - L_t - C_t \\
E_{t,below} &= -C_t && \text{since } \beta \cdot C_t > 0
\end{aligned}
\tag{14}
$$

**Assumption 2:** Linear utility function is assumed which implies no aversion to risk,

$$
\mu(E) = E
\tag{15}
$$

Substituting Eq. (14) into Eq. (4), applying the linear utility function assumption, and simplifying for only two possible states using $p$, the forecast probability of flow above the flow threshold, leads to.

$$
\begin{aligned}
U(\tilde{E}_t) &= p_t \cdot E_t^{above} + (1 - p_t) \cdot E_t^{below} \\
&= p_t \cdot \left[\min\left(\beta \cdot C_t, L_t\right) - L_t - C_t\right] + (1 - p_t) \cdot \left[-C_t\right]
\end{aligned}
\tag{16}
$$

**Assumption 3:** Probability of flow above the threshold will always be either 1 or 0,

$$
p_t \in \{0, 1\}
\tag{17}
$$

We can now determine the single timestep ex ante utility for the four possible outcomes; forecast probability is 1 or 0, and an action has been taken or not, leading to Table 4.

**Table 4: Ex ante utility values for a time-step of Expected Utility Theory with REV assumptions**

| | $p = 1$ | $p = 0$ |
|---|---|---|
| Action taken $C_t \neq 0$ | $-\left(C_t + L_t^u\right)$ | $-C_t$ |



| | Event occurred | Event did not occur |
|---|---|---|
| Action not taken $C_t = 0$ | $-L_t$ | 0 |

Applying Eq. (8) to Eq. (16) will lead to an optimal amount $C_t^\pi$ to spend on the mitigating action for each timestep. By considering that the probability is always either 1 or 0 due to assumption 3 and that all costs and losses are positive values

680   we can derive that for any timestep the cost will be either $C_t^\pi = 0$ when $p = 0$ or $C_t^\pi = \dfrac{L_t}{\beta}$ when $p = 1$.

The ex post utility for each timestep, shown in Table 5, can be found by substituting these optimal costs back into the elements of Table 4, and letting the probability be conditioned on the state of observed flow above the threshold.

**Table 5: Ex post utility values for a time-step of Expected Utility Theory with REV assumptions**

| | Event occurred $p_t = 1$ | Event did not occur $p_t = 0$ |
|---|---|---|
| Action taken $C_t = \dfrac{L_t}{\beta}$ | $-\left(\dfrac{L_t}{\beta} + L_t^u\right)$ | $-\dfrac{L_t}{\beta}$ |
| Action not taken $C_t = 0$ | $-L_t$ | 0 |

685

A contingency table is now used with Table 5 to determine each term of the RUV metric.

**Assumption 4:** The frequency of the binary decision event $\overline{o}$ is used for the reference baseline.

This leads to the following expected ex post utility for reference information

$$\mathop{\mathbb{E}}_{t \in T}\left[\Upsilon\left(E_t^r\right)\right] = -\min\left\{\frac{L_t}{\beta}, \overline{o}L_t^a\right\} - \overline{o}L_t^u \qquad (18)$$

690   Expected ex post utility for perfect information is

$$\mathop{\mathbb{E}}_{t \in T}\left[\Upsilon\left(E_t^p\right)\right] = -\overline{o}\left(\frac{L_t}{\beta} + L_t^u\right) \qquad (19)$$

Expected ex post utility for forecast information is

$$\mathop{\mathbb{E}}_{t \in T}\left[\Upsilon\left(E_t^f\right)\right] = -(h+f)\frac{L_t}{\beta} - \overline{o}L_t^u - mL_t^a \qquad (20)$$

where $h$ is the hit rate, $m$ is the miss rate, and $f$ is the false alarm rate from the contingency table.

695   **Assumption 5**: At each timestep the avoided losses are equal to the total possible losses.



$$L_t = L_t^a \qquad \text{for } t \in T \tag{21}$$

Substituting Eq. (18), (19), and (20) into Eq. (10), applying assumption 5, and noting the relationship $\beta = \dfrac{1}{\alpha}$ leads to

$$RUV = \frac{\min(\alpha, \overline{o}) - (h + f)\alpha - m}{\min(\alpha, \overline{o}) - \overline{o}\alpha} \tag{22}$$

which is identical to the definition of the REV metric in Eq. (2).

**Data availability**

A companion dataset for this work is available at: https://doi.org/XXX. This contains the input streamflow forecasts, output forecast value results, and generated figures. The software library used to generate the forecast value results is not included in this dataset because it will be released with a follow up publication.

**Author contributions**

Richard Laugesen led the conceptualisation, data curation, formal analysis, funding acquisition, investigation, methodology, project administration, resources, software, supervision, validation, visualisation, writing – original draft preparation, and writing – review & editing. Mark Thyer supported funding acquisition, investigation, methodology, project administration, supervision, visualisation, and writing – review & editing. David McInerney supported formal analysis, methodology, resources, visualisation, and writing – review & editing. Dmitri Kavetski supported methodology, visualisation, and writing – review & editing.

**Competing interests**

The authors declare that they have no conflict of interest.

**Acknowledgements.**

This work was conducted on the traditional lands of the Ngunnawal people and Kaurna people. We acknowledge their continuing custodianship of these lands and the rivers that flow through them, and pay our respects to their elders, past and present. We also acknowledge the traditional custodians of the catchments and rivers used in this study. The authors thank Beth Ebert, Michael Foley, and Prasantha Hapuarachchi for their review of this paper and thoughtful discussions on the method, and Jacqui Hickey for her formative discussions on the use of forecasts for operational decision making at the MDBA and encouragement to pursue this topic. Richard Laugesen is grateful to the Bureau of Meteorology for their



generous support of his research, particularly Narendra Tuteja, Alex Cornish, and Adam Smith for seeing the value of this approach. This work was supported through an Australian Government Research Training Program Scholarship and with supercomputing resources provided by the Phoenix HPC service at the University of Adelaide.

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
