# Peer review of "Flexible forecast value metric suitable for a wide range of decisions: application using probabilistic subseasonal streamflow forecasts"

_Hydrology and Earth System Sciences, 2022_

## Author Comment (AC1)

**Response to comments from Reviewer 1**

**Summary**

1.1.   The manuscript by Laugesen et al. introduces a new metric to assess forecast value adapting the formulation of a previously existing metric, namely Relative Economic Value, within a flexible value assessment framework based on utility. The method is then exemplified with subseasonal forecasts in the case of the Murray River, Australia, where decisions tend to target high flow values. A sensitivity analysis is carried in this case study.

The paper, which proposes a new methodology and results of high significance for the forecasting community, is detailed, very didactic and of high quality, and will undeniably be valuable to researchers who wish to carry out advanced and flexible forecast value analyses, involving decision-makers' levels of risk aversion.

I strongly recommend this paper for publication and list hereafter recommendations for clarification, as well as some minor points and typos.

We thank Reviewer 1 for their encouraging feedback and detailed review of our paper. In particular, we appreciate their thoughtful suggestions for improving the clarification of certain sections, which will make this material easier to follow.

**Comments**

1.2.   L18-21: These two sentences seem a bit contradicting because you first announce value for all lead times, decision types and most levels of risk aversion, but then you nuance your statement beyond the second week, for binary decisions. I suggest nuancing the first statement.

We agree that these two sentences appear contradictory, and will adjust our wording to rectify this problem.

1.3.   In addition, the case of the Murray-Darling basin being an example of application for sensitivity analysis rather than a stand-alone evaluation, I would consider these results as secondary compared to the advantages of the proposed RUV metric and the results of the sensitivity analysis well described in Section 6.2, which in themselves deserve to be highlighted in the abstract.

Good point. We agree that the Murray Darling experiments are an example of a sensitivity analysis, more so than an evaluation of forecasts, and will revise our text to reflect this. In particular, we will modify the abstract to highlight the outcomes of the sensitivity analysis in section 6.2, choose a new term for the "case study", and make the "case study" results secondary.

1.4.   L20: "Beyond the second week" please mention that you are referring the lead time.

Good suggestion. We will implement this clarification to avoid potential confusion.

1.5. L26 (and throughout the paper): Here authors refer to the lens of "consumer" impact. The terms "user" and "decision-maker" are also used throughout the paper. Given that there are differences between these terms, I wonder whether the authors could clarify whether they use these three terms as interchangeable, or do they make a distinction. In the former case, are they actually interchangeable? Forecast datasets are increasingly open, and I am not sure whether users are indeed consumers in these cases. In the latter case, could you explicit the distinction made in an evaluation context?

This is a good point. We agree that these terms were not used consistently and will correct this by using the term "user" through the paper. We feel it is important to be clear that the "user" in this context is an individual making a decision and therefore we will replace "decision-makers" on line 11 with "decision-makers (users)".

1.6. L73-78: Based on these two examples, and purely intuitively, I would tend to consider both types of decision-makers to be risk averse (conservative approach to avoid spending in example 1 and flooding in example 2) but with a different sensitivity to forecast uncertainty. Could the authors elaborate on why they make a direct link between forecast uncertainty and risk aversion?

Thank you for raising this important point of clarification. There is potential for confusion, as the formal definition of "risk aversion" does not always align with the colloquial interpretation used in daily life. In our study we use the term "risk aversion" from the economic literature, as defined on lines 73-74. However, we see that this definition is not entirely clear, and we will improve this with specific mention to the relationship to uncertainty and cite (Mas-Colell, 1995). We will also remove example 2 from the paper, as we now appreciate that it is unnecessary and complicates the introduction of the "risk aversion" concept.

1.7. L95: Maybe reformulate "lead to improved forecast verification". For instance: "lead to improved forecast verification indicators" or "improved forecast performance".

Good suggestion. We will change this wording.

1.8. L98: "first convert them"

Thanks. We will correct this.

1.9. L125: Isn't it a 2x2 contingency matrix?

Yes it is. We will fix this typo.

1.10. L133: The term "outcome" was unclear to me here. I was unsure whether it referred to each combination of possible Action/Event in Table 1. In my understanding, E depends on each information source (reference, forecast, or perfect) but uses all possible outcomes in its weighted mean. The term "outcome" was a bit confusing, while Equation 1 and L138 were perfectly clear. Since the Supplement helped in that matter, I would suggest referring it here already.

Thanks. We will add a sentence defining "outcome", and also refer to the supplement.

1.11. Equation 1: Could you please add the range within which V should fall (-∞ to 1)?

Sure, that is a great suggestion.

1.12. Equation 2: At this stage o is not defined.

Thanks for catching this oversight. We will define $\bar{o}$ as the frequency of the binary decision event (as on line 687) in the sentence following equation 2.

1.13. Figure 1: The location of the phrase "Use reference to decide" is, I think, misleading. Based on the explanations (L162-164), it seems that for a cost-loss ratio of 0.5, for instance, the forecast outperforms climatology and should thus be used to decide, with a potential REV reaching about 0.8. Therefore, using the reference for a cost-loss ratio of 0.5 would not allow reaching a REV greater than that of the forecast. However, based on the figure, it seems that using the reference for a cost-loss ratio of 0.5 would allow reaching a REV greater than that of the forecast. Maybe the arrows pointing at the extreme intervals when the reference is indeed performing better, but this is currently not clear.

We see how the position of the "use reference to decide" text could introduce confusion. Our intent was as follows, the sentence "use reference to decide" is part of the "always act" and "never act" arrows, as in "use reference to decide and always act" and "use reference to decide and never act".

We will replace the annotations on the figure to clearly label the different regions of the value diagram using (a), (b), and (c) and add concise text on how to interpret each:

(a) $0<=\alpha<0.05$ – use reference to decide, and always take mitigating action

(b) $0.05<=\alpha<0.95$ – use forecast to decide whether to take mitigating action

(c) 0.95<=$\alpha$<1 – use reference to decide, and never take mitigating action

We believe that suggested revisions to the figure will make this clear.

1.14. Additionally, it is not clear whether the arrows linked to "Always act" and "Never act" point at the interval when climatology < forecast or at the specific points (0;0) and (0;1) (see also the following response to comment 1.15).

Thanks for highlighting this. Our intention was for the arrows to referring to the intervals rather than (0,0) and (0,1). We will adjust the figure to make this clearer.

1.15. Figure 1 (and all value diagrams): If I understand correctly the meaning of $\alpha$=1 (never worth acting) and $\alpha$=0 (always worth acting), the decision can be taken regardless of whether the forecast or climatological information is considered. This would mean that the relative economic value should be exactly equal to 0 in both cases ($\alpha$=1 and $\alpha$=0). If that is correct, and that no other parameter comes into the decision of acting or not, is there a reason why the two points (0;0) and (0;1) are not represented in the value diagram?

Thanks for this comment. The above interpretation of the cases $\alpha$=1 and $\alpha$=0 is not quite correct. There is no conceptual reason why the relative value should be zero at these end points as this would imply that the forecast and reference climatology are both equally valuable, which is unlikely. In our illustrative example the reference climatology is more valuable.

REV uses a fixed average value for the reference climatology and there are only two possible actions:

1. Always act (when $\alpha < \bar{o}$ ) or
2. Never act (when $\alpha >= \bar{o}$ )

(where $\bar{o}$ is the observed event frequency).

This is detailed in the derivation on lines 30-34 of the supplement.

A decision-maker *should* use climatology to make decisions when climatology is more valuable than forecasts, and therefore they will use one of these two options if their $\alpha$ value lies in intervals the arrows are pointing at in our illustrative example.

We will add a reference to the derivation in the supplement, and cite (Richardson, 2000) which introduces REV, the value diagram, and its interpretation.

1.16. Equation 4 (and Equation 9): Probabilities being sometimes used with powers, I would suggest to place the index m as a subscript rather than superscript.

Thanks for pointing this out. We will use subscripts for the *m* index on probabilities rather than superscripts.

1.17. L207-217: I suggest adding an example graph of µ to illustrate your explanation. For instance, I find it hard to picture the concavity of µ, especially in the case of binary decisions.

Thank you for this suggestion. We will consider the benefit of adding a second panel to figure 3 with a graph of $\mu$ for the 4 values of $A$ used in the study. We will also add an explanatory sentence to the risk aversion section 4.6.

1.18. L230: "the absolute value of a specific decision"

Thank you for picking this up. We will fix this typo.

1.19. Equation 6: Here it is not clear to me why damage does not vary with time (in Appendix A it seems it does). It is also not clear why m, whom E, b and d depend on, appear in parenthesis in the case of b and d, and as a subscript in the case of E.

We agree that there is inconsistency in our notation, especially with the way we are indexing $t$ and $m$.

The reviewer is correct that observed damages vary with time, as the realised state of the world associated with each observation changes with time. We will make this clear in our revised notation for equation 6, and across the whole paper, appendix, and supplement.

1.20. L237 "The damage function relates the streamflow magnitude to the economic damages": At this stage, you have not mentioned streamflow yet, I would suggest sticking to the term "states of the world".

We agree that would be clearer. Thanks!

1.21. L309: The previous section also comprised elements of methodology. Consider changing the name of this section.

Good suggestion. We will change this section title to a more appropriate term that describes the content.

1.22. Section 4.2: Could you briefly state why you chose this station and basin?

Good idea. We will explain that this site is significant for water resource management because it is upstream of a major water storage.

1.23. To which extent do you expect your results (sensitivities) to differ in a catchment with different hydrometeorological characteristics?

We will mention that the results are likely to be sensitive to the flow characteristics and forecast uncertainty and that other sites will be analysed in future work.

1.24. L340-341: Given that you mention a rainfall post-processing step, I would recommend stating "raw streamflow forecasts" (L340) and "the streamflow observations" (L341) to avoid any misunderstanding.

Thanks. This change will avoid potential confusion.

1.25. Section 4.3: GR4J also uses temperature or potential evapotranspiration as input. Could you say something about what you used?

Good catch. We will add that PET from the AWAP model has been used.

1.26. L345-346 "flow exceeding the height of a levee": it would be more intuitive to talk about the "water level exceeding the height of a levee"

We agree, and will make that change. Thanks.

1.27. L374: "all decision-makers share the same level of risk."

Thanks. We will fix this typo.

1.28. Table 3: (1) "Experiment 4: Impact of risk aversion on forecast value"; (2) In experiment 5, the decision thresholds says "All flow" but the decision type is "Binary", which is counter-intuitive. "All possible thresholds" might be easier to understand, or "Thresholds from bottom 2% to top 0.04%".

Good suggestion. This will make it easier to understand.

1.29. Figure 4: Here you consider two rather extreme yet probably realistic thresholds for converting the probabilistic forecast into a deterministic one. When reading the results, I was wondering whether moderate thresholds could alleviate the lack of forecast value for

high and low cost-loss ratios and provide reasonable value for all cost-loss ratios. Could you answer this by displaying intermediate probability thresholds in this experiment?

Thank you for suggesting this. We will add an additional intermediate probability threshold to further illustrate this issue.

1.30. Figure 6: To ease the reading of this figure whose lines are plain and with colors of similar intensity, I suggest adding dashes and dots to distinguish the three curves.

Good suggestion. We will add additional line styles to make it easier to read the figure and ensure the colour choices are colour blind friendly.

1.31. Figure 6: Could the authors explain the interesting difference in RUV pattern for the multi-categorical decision (also seen in other decision types) between lead week 2 and lead weeks 3 and 4? Why does value decrease with lead time for low cost-loss ratios (as expected) but increases with lead time (maybe less obvious) for high cost-loss ratios?

Thank you for bringing this pattern to our attention. While the differences between weeks 2 and 3/4 are minor they interestingly appear robust to decision-type. We will provide some possible reasons for these differences, namely lead-time dependent differences in the post-processor correction of forecast errors in low/high flow regimes, and decreasing sharpness of the forecast ensemble at longer lead-times. A definitive explanation would require a dedicated experiment and will be sought in future work that focuses on a decision-maker application and/or additional forecast locations. We will mention this future research direction explicitly in Section 6.3 on future work.

1.32. Experiment 3: In this experiment, authors look at the variation of value with the lead week. It is also common to look at the influence of the initialization month or season to appreciate the influence of different hydrological conditions on the value. Even though it would mean dividing the total forecast sample into subgroups and reducing significance, I think it could be a valuable addition to Figure 6 to show forecasts initialized in dry and wet conditions separately.

Yes, this is a great point and we agree that assessing the impact of seasonality and antecedent conditions on forecast value is an important research question. While important, we believe this assessment does not fit is not within the research aims of the current paper, which focus on the introduction of RUV and a comparison to REV.

The Biggara case study is used as a vehicle to illustrate the general application of the RUV method. The current set of case study analyses is already quite extensive and includes results from 5 experiments illustrated through 8 figures. To sufficiently assess the impact of different hydrological conditions on forecast value we would need to add an additional

experiment with dedicated figures and text to explain the impact of the seasonality and antecedent conditions with respect to the decision-types and lead-times. Further, for completeness the impact should also be assessed on the risk aversion outcomes through additional evaluation in experiments 4 and 5. We feel that this additional content would distract from the main aims of the paper and result in an unduly lengthy manuscript.

After some reflection, we feel this interesting and important research question requires a dedicated separate study and is best left for future work. Thank you for raising this important issue for practical application. We will add text to Section 6.3 that addresses this and outlines further work is needed.

1.33. Figure 7: To ease reading, consider adding a horizontal line at y=0 in graphs displaying the overspend.

This is a good suggestion. We will do this.

1.34. Experiment 4: It is currently unclear why the third line of Figure 7 is shown as it is little to not exploited in the interpretation. Please consider removing or spending some sentences to exploit this line of the figure.

Thank you for this suggestion. We assume you are referring to the third row of panels rather than third line. One intent of including the utility-difference (and overspend) results was to enable a more direct comparison of our findings with those in Matte et al. (2017), line 469-471. We agree that adding an interpretation of the utility-difference results would add value and thank the reviewer for pointing out this oversight. We will add additional text interpreting the utility-difference results at line 485.

1.35. L520: "making decisions with fixed critical probability thresholds leads to"

Good catch. We will fix this.

1.36. Sections 6.1, 6.2 and 6.3: Numbering the paragraphs is unnecessary.

Thank you for highlighting this. We will remove the paragraph numbering.

1.37. L576: "summarizes"/"summarises"

We will correct this Australian/US language issue, and check the rest of the document.

1.38. L680 and Table 5: In the text, you mention that the formulation of Ct depends on the value of p, but in Table 5, the formulation of Ct depends on whether the action is taken or not rather than on p. I could not figure out why. Are the p values you are referring to in both instances different? Could you please clarify this point?

Thank you for pointing this out. We agree that it is not clear from the text in the appendix. Section 2 in the supplement includes a more complete derivation but on reflection that is also lacking clarity.

The probability on line 680 is referring to the forecast probability of flow above the threshold, which then determines ex ante (i.e. before event has taken place) how much to spend on the action $C_t^\pi$. The probability in Table 5 is referring to the ex post probability that the observed flow is above the threshold. The amounts spent on action or no action in the row titles of Table 5 are the optimal costs $C_t^\pi$ found ex ante (i.e. after event has taken place).

We will make the following changes to the main text and the supplement to improve the clarity of this derivation:

- On line 679 replace "probability is always 1 or 0" with "forecast probability is always 1 or 0"
- Change the column titles of Table 4 and Table S2 in the supplement to "Event forecast to occur" and "Event forecast to not occur".
- Change the column titles of Table 5 and Table S3 in the supplement to "Event occurred" and "Event did not occur".
- On line 673 replace "forecast probability is 1 or 0" with "event is forecast to occur (p=1) or not occur (p=0).
- On line 118 of the supplement replace "letting the probability be conditioned on observed flow above the threshold" with "letting the probability be conditioned on observed flow above the threshold, rather than the forecast flow used for the ex ante utility".

1.39. L701: The link to the companion dataset is missing.

Thanks for pointing this out. We will include a permanent link to the companion dataset in the revised manuscript.

**References**

Mas-Colell, A. (1995). *Microeconomic theory*. Oxford University Press.

Matte, S., Boucher, M.-A., Boucher, V., & Fortier Filion, T.-C. (2017). Moving beyond the cost–loss ratio: economic assessment of streamflow forecasts for a risk-averse decision maker. *Hydrology and Earth System Sciences*, *21*(6), 2967-2986. https://doi.org/10.5194/hess-21-2967-2017

Richardson, D. S. (2000). Skill and relative economic value of the ECMWF ensemble prediction system. *Quarterly Journal of the Royal Meteorological Society*, *126*(563), 649-667. https://doi.org/10.1002/qj.49712656313

---

## Author Comment (AC2)

**Summary**

2.1.   The paper presents a generalisation of the Relative Economic Value (REV) approach, providing a flexible metric, the "Relative Utility Value" (RUV), which can be useful for decision makers on the value of probabilistic subseasonal forecasts. The results show its application and sensitivity to several factors in a case study in Australia.

The paper is well written and demonstrated. I believe it brings novel aspects in the topic of hydrometeorological forecasting, and is an excellent demonstration of how forecast producers and users should work together to enhance the usefulness of skilful forecasts.

I have just some minor general and specific comments, presented below.

Thank you for this thorough, well considered, and detailed review. We appreciate your words of encouragement and suggestions which will improve the quality of this work.

**General comments:**

2.2.   I think some sentences need to be more carefully revised because they might convey a message that goes beyond the experimentations of this paper. For instance, concerning the first sentence of the Conclusions section, I do not believe that, overall, the value of probabilistic forecasts to making (good) decisions has not been established, as the authors say. Many public and private companies are convinced of the value of quantifying uncertainties in real-time forecasting and that is why this type of forecasts has been increasingly produced and used for many operations, from nowcasting to short-term flood forecasting and long-term inflows to reservoirs. Value has not been established (or explicitly calculated) at all lead times and users cases, I agree, but, overall, the forecasting (producers and users) community acknowledges that there is value for decision making in not being certain (or deterministic) about the unknown future. The added value of the paper, in my opinion, does not lie on bringing the "value" into discussion in forecast verification/evaluation, as this has been done in several papers previously, but in making the framework for assessing it more accessible and flexible, as the title says.

Thank you for bringing this important point to our attention. We now recognise that some statements in our paper are too general, and go beyond the experimentations in our paper. In particular, we will rewrite parts of the Conclusions to address this problem, including:

- Line 626-678: Replace "but to date their value for decision making has not been established. Forecast value methods attempt to quantify this potential" with "and forecast value methods attempt to quantify this potential.".

- Line 633: Replace "can be incorporated by the user" with "can be incorporated into RUV by the user".

- Line 638: Replace "decision-maker characteristics" with "user risk aversion and exposure to damages".

Our proposed changes in response to the following comment (point 2.3) will also align the message with the experimental results.

We also agree that the main contribution of this study is to introduce a more flexible framework for quantifying forecast value, rather than to establish the value of probabilistic forecasts in all cases, or their value over deterministic forecasts. We note that this is stated in the research aims on lines 104-108 and supported in the Introduction at lines 92-97.

2.3. I was also puzzled by the authors when they say that a decision maker who is highly exposed to damages should use the reference climatology rather than a forecast based on meteorological numerical models for binary decisions (Conclusions, lines 639-640). This might be the case for the experiment showed (and the case described in the paper), but I doubt flood forecasters (forecasting a threshold exceedance for the next 12-24 hours, for instance) would be able to say to the population they are serving that they will abandon a city located close to a river and leave than with only a climatology-based information instead of rather investing into a (good) model-based forecasting and alert system because they are highly exposed to damages. I fully understand that if the potential costs of a flood event are high, and will be incurred if the flood occurs, whatever forecast we might deliver, then no forecasting system can save us, and it is better to work on protection (decreasing costs) at first. But even in this case, using climatology might not be beneficial either (the problem is elsewhere, not in the type of forecast being used). What I mean is that out of a more explicitly presented context, some sentences might rather diverge a reader from the purposes of the paper. Therefore, I would recommend to revise some general affirmative sentences, or at least introduce more context to them to avoid misunderstandings.

Thank you for this considered comment. Reviewer 1 (point 1.3) also asks that we revise the conclusions to focus more on the outcomes of the sensitivity analysis experiments, rather than outcomes of the case study. When we re-write the conclusions, we will avoid general affirmative sentences and state that the outcomes are context dependent and provide clear information on that context.

For example, we will adjust lines 636-641 to state the case study context and outcomes more explicitly, from:

"A case study demonstrates that subseasonal streamflow forecasts should be preferred over a reference climatology forecast for all lead-times studied (max 30 days) and almost all decision-makers regardless of their risk aversion. This positive forecast value is robust to changes in decision-maker characteristics, decision types (binary, multi-categorical, and continuous-flow), and decision-making approaches. However, beyond the second week, RUV indicates that decision-makers who are highly exposed to damages should use the reference climatology rather than the forecasts for the binary decision. This is not the case for the multi-categorical and continuous-flow decision however, where forecasts should be preferred."

to:

> "A case study using a cost-loss economic model at Biggara in the Southern Murray-Darling Basin of Australia assessed the relative value of subseasonal streamflow forecasts over a fixed historical average reference climatology. This case study demonstrates that the forecasts should be preferred over the reference climatology forecast for all lead-times studied (max 30 days) and almost all users regardless of their risk aversion. This positive forecast value is robust to changes in user risk aversion, decision types (binary, multi-categorical, and continuous-flow), and decision-making approaches. However, the results indicate that users who are highly exposed to damages would gain more value using the reference climatology rather than forecasts for the binary decision in lead-time weeks 2-4. This was not the case for the multi-categorical and continuous-flow decision however, where the forecasts should be preferred. As REV is limited to binary decisions, a user making a multi-categorical or continuous-flow decision, could be misled by the REV outcomes and consider not using the forecasts when they actually have significant value as demonstrated by RUV."

2.4. a) Another general comment is about the fact that we set the context of the paper on probabilistic subseasonal forecasts (up to 30 days), but much of the demonstrations and experiments refer to 1-7 day lead-time forecasts (and many concluding sentences seem to forget this context and generalize to any type of forecast and lead time).

Thank you for highlighting this. During preliminary investigations we did generate results for other lead-times and groupings. The specific case-study features used (as described in section 4) were selected to best present the salient features of RUV and the case study results. This is described on lines 311-313. On reflection, we agree with the reviewer that some concluding statements have been generalized beyond the experimental results shown in Section 5. We will rectify this when re-writing the conclusions. Please see related responses to comments 2.2 and 2.3.

b) In many situations (but I am not sure about the case of the particular catchment of the study), a meteorological (model-based) forecast may show quality a couple of days ahead (1 to 5 days, for instance) and then be as skilful as climatology afterwards. How this difference in the quality of the forecasts might affect the results here? Is it justified to group together these lead times here?

This is an interesting point. We agree that rainfall forecasts are only skilful for short lead times (e.g. 1-5 days). However, streamflow forecasts are typically skilful at longer lead-times than rainfall forecasts due to storage effects. In particular, the subseasonal streamflow forecasts used in this case study have previously been shown to be sharper and had higher CRPS skill scores than climatology for lead times up to 30 days (see McInerney et al. (2020) for a detailed explanation and evaluation of this. Additionally,

Figure 6 demonstrates that these forecasts have higher value than climatology for longer lead-times.

We found that grouping lead times together in our analysis was beneficial in addressing the aims of this paper, namely to introduce RUV and contrast it with REV. We also found that the particular groups (1 week, 2 weeks, 3-4 weeks) demonstrated key differences between lead times.

However, grouping lead-times would not be recommended in a practical context. If the purpose of this case study was to quantify value of using forecasts to inform river operations, then we would need to analyse forecast value of individual lead-times. We will acknowledge that grouping lead-times is not recommended for practical applications in section 4.7 when we introduce the groupings. Please see our response to a related comment (1.31) from reviewer 1.

c) Would a (potential) difference in quality explain negative RUV (lines 412-414), where the authors say that climatology (as a forecast) is more useful than a (meteorological model based) forecast?

We do not believe that the differences in forecast performance across the 1st week are the root cause of negative value regions discussed on line 412-413. The negative regions are a consequence of using a fixed critical probability threshold, as explained in section 5.1 and the REV literature, see cited references Richardson (2000) and Murphy (1977). This result is seen for any forecast regardless of whether lead-times are grouped or not. We will add an additional sentence at line 417 stating the generality of this result with reference to the REV literature.

d) (note: at the end, the decision maker is always using a forecast, either from a record of historic observations – climatology – or from a coupled atmospheric-hydrologic model).

This is of course true. However typically the term "forecast" refers to a procedure more complex than "just" the marginal distribution of historical data.

Thank you for bringing this need for more clarification to our attention. We agree that the term "forecast" typically refers to a reasonably complex procedure. As this study involves a comparison of the new method RUV to the existing method REV we are limited to using the baseline reference used by the REV. This critical limitation of REV is stated on lines 65-66 of the Introduction and is a key motivation for the development of RUV. We will include the need to apply RUV with "practically relevant reference forecasts" in Section 6.3. Please see our response to a related comment (1.32) from reviewer 1.

2.5. Finally, a last overall comment I have is: why a systematic comparison with REV is so important in the development of a novel approach or metric in this topic? Is it because

REV is widely used (or supposedly widely used)? How crucial is it as motivation for the study?

The reviewer is correct that the comparison with REV is important because REV is commonly used. We feel this is an important motivation for the study, and make the following points about this in our paper:

- REV is widely used in the literature for quantifying the value of forecasts (see lines 54-61),

- The primary way to present this information is using a value diagram (see lines 89-90).

- As the value diagram is a compelling way to communicate value across a range of decision-makers, and the community is familiar with it, we felt it was important to be able to leverage this with any new approach (see lines 265-266).

**Specific comments:**

2.6.  Introduction: I think the authors could introduce some literature on works done on forecast value and links between forecast quality and value with respect to inflows to hydropower reservoirs. These cover a large range of cases and lead times, and also use optimisation-based economic models to link forecast production (quality) to usefulness (economic value). It would be interesting to give here this broader view to the topic, I think, and then replace better the context of the paper (to which the conclusions drawn will specifically apply). Besides the paper mentioned in the discussion (Penuela et al.), some others that might be interesting are: https://doi.org/10.1002/2015WR017864; https://hess.copernicus.org/articles/23/2735/2019/; https://doi.org/10.1029/2019WR025280; https://hess.copernicus.org/articles/25/1033/2021/.

Thank you for this suggestion and for providing these references. Originally, we did not want to add context to the introduction beyond forecast value and the case study, and instead left broader context to the discussion. Your reasoning has convinced us to include some text on the links between forecast quality and value, and the optimisation-based modelling in hydropower. Thanks

2.7.  Line 49: too many "and" words. Please, check.

We do not see any "and" words on line 49 and feel the sentence reads fine.

2.8.  Line 50: "better verification implies more value": I think you refer to "quality" and not "verification". Please, check.

Correct. Thanks for catching this.

2.9.   Line 88-89: not clear to me. Please, check.

Thanks. We will rewrite this sentence for clarity.

2.10.  Line 90, 102: when you refer to "the authors" I am sometimes a bit confused if you mean "you" or the authors in Matte et al. Please, check.

Thanks for noting this. We will replace "the authors" with "we" in these two sentences.

2.11.  Line 192-193: maybe it is not reported in scientific papers, but are you sure it is not commonly used by water managers in practice? Have you conducted a survey or any other study not reported here to assess it (i.e., real-world practices)?

Good point. This statement was based on our knowledge of the scientific literature and 15 years professional experience providing forecasts to water managers in Australia. However, we recognise that this statement was too broad, and will soften the language accordingly. In particular, we will mention that, to the best of our knowledge, making real-work decisions with $p_t = \alpha$ has not been reported in the published literature.

2.12.  Line 227-230: again too many "and" words. I found the sentence unclear. Please, check (maybe also correct to "a specific decision").

Thanks for spotting this. We will remove the unnecessary "and" from line 227. We assume that you are referring to the sentence on lines 228-230 as being unclear. We agree and will split this into two sentences, as well as adding a little more context.

2.13.  Line 280: I am not fully convinced that information on amount spent, damage etc. at each time step is something valuable to a user. Is that so? Can you provide examples or a justification for that? I believe that users might be more interested in the long-term performance of a forecast system (in particular when it comes to reservoir operations), while a flood alert user would be interested in the whole flood event duration performance (and less on each time step). Maybe I misunderstood something here.

We will provide an example of users who may find this additional information useful.

One example is a user applying alternative economic models or tuning damage functions to match real-world data. This user would require the amount spent and damages incurred at individual time steps to determine that the model is behaving as expected.

A second example is a decision-maker who has finite funds to spend on the mitigation of damages. Understanding when the maximum spend amount exceeds the funds available would require investigation of spend and damage amounts at individual time-steps.

2.14. Line 309: I do not think "Methodology" is a good title for the section. I would suggest "Application" or "Experiment".

We agree with this suggestion, and will change this heading to better describe the content.

2.15. Line 310-311: I guess that by "different decision-makers" you mean "different levels of aversion of decision-makers". I think it is not the person themselves you are talking about but the theoretical level of aversion that you are modifying in the experiments.

Correct. We are referring to the level of risk aversion of an individual decision-maker, and their exposure to damages ($\alpha$). We will reword "decision-makers" to "decision-makers with different exposure to damages and different levels of risk aversion".

Thank you for bringing this to our attention.

2.16. Section 4.1: I think part of it could go to the Introduction.

Thank you for this suggestion. We agree that some of this material could be included in the Introduction. The intent for Section 4.1 is to provide background and motivation which is specific to the case study application introduced in Section 4. Therefore, we will move lines 315-320 to the Introduction and adjust the remaining sentences for clarity.

2.17. Line 339: maybe place the references in the right place would help the reader (ex. Perrin et al., after GR4J, and not after RRP-S).

Thank you for this suggestion. We will move the references from the end of the sentence and place after each modelling component.

2.18. Line 343: "seamless" has usually another meaning in the literature. It usually refers to a system that forecasts in a coherent and homogeneous way from minutes to hours and months. It is not usually related to performance across scales. Please, check.

In this context, "seamless" refers to forecasts which are coherent and homogenous and have similar forecast quality across time scales. We will rephrase the sentence on line 343-344 sentence to clarify how we are using the term.

2.19. Section 4.4: I think part of it could go to the Introduction (lines 346-354).

Thank you for this suggestion. We agree that the Introduction does not provide enough context or motivation for the different decision types. While lines 53-54 briefly introduce the decision-types we feel that having more context up front would help the reader and will add additional sentences to the Introduction based Section 4.4 (lines 346-354).

2.20. Line 369: what do you mean by "suitable"? How? Based on data?

Good point. We did not define what we meant by "suitable". We will clarify that it reproduced the real-world assumptions described in the previous sentence.

2.21. Table 3, experiment 4: check typo

Well spotted. We will remove the "of" typo

2.22. Fig. 4: I am not sure it is needed to show that we come up to the same results.

Although the value diagrams in panel (a) and (b) are identical, they are not the same thing. They are the same outcome from two different methods. We feel it is important to show this equivalence in an explicit way as it supports the research aims of this study. However, on reflection we can see how it may have caused confusion and will change the caption of panel (b) to "RUV with restrictive assumptions equivalent to REV" to make the difference clearer.

2.23. I would suggest putting Experiment 1 and Experiment 2 together.

Thank you for this suggestion.

Although the results shown in Figures 4 and 5 are similar, they are different experiments with different reasoning and explanations. We prefer to keep the two experiments distinct for clarity and introduce the content in a staged way. We agree that combining the results

together as a single figure would be more efficient in terms of space; however, it would be more difficult to explain and describe the differences. By keeping the results separate it is easier for the reader to understand the reasons for the similarity and differences between REV and RUV, which is a research aim of the paper.

2.24. Line 437: what do you mean by "ensemble sampling error"? Please, explain.

Thank you for noting this. We were referring to errors due to the small ensemble size. We will replace "ensemble sampling error" with "sampling errors due to small ensemble size".

2.25. Line 458: please, clarify the sentence (see my general comments above) in terms of saying that a "decision-maker should avoid using forecasts" in certain conditions.

Thanks for pointing this out. This point is discussed in our response to reviewer comment 2.3.

2.26. Line 464-466: Does this correspond to reality? Have you discussed the results with the Murray-Darling Basin managers, for instance? It would be interesting to link mathematical calculations to reality in the field, providing supporting to some sentences on the results and overall conclusions drawn in the paper.

We agree that it would be interesting to make a strong link between forecast value methods such as RUV and applications "in the field".

As pointed out by reviewer 1, this paper is not aimed at providing a detailed and conclusive evaluation of forecast value for Murray Darling Basin managers. It is aimed at introducing RUV and contrasting it to REV by demonstrating that RUV can handle factors which are important to real-world decision making. This is demonstrated through experiments showing the conditions under which RUV and REV are equivalent and illustrating the sensitivity of forecast value to different decision types (binary, multi-categorical, and continuous flow) and levels of risk aversion. For the case study in this paper, we have made reasonable assumptions about the damage functions and decision thresholds, and necessarily used the cost-loss economic model for comparison with REV. We feel this is sufficient for the purposes of this paper, which is to demonstrate that forecast value is sensitive to decision-type and levels of risk aversion.

Now that we have established forecast value is sensitive to these choices, in the future we can undertake a realistic evaluation of forecast value for Murray Darling Basin managers using RUV. We will add this topic to the future work section, mentioning the need to "calibrate" the damage function, decision thresholds, and economic model based on the real-world experience of decision-makers.

2.27. Fig. 7: I think it should be more commented. The differences we see in the column on the right do not seem to be "moderate".

We will expand the commentary on lines 471-473 and lines 486-487 to ensure it is clear that the impact is "moderate", except for the case of highly risk aversion decision-makers with continuous flow. Thanks for pointing this out.

2.28. Experiment 5: could you justify the choice of adopting a binary decision and alpha = 0.2 here? Also, why are you showing week 1 if the focus of the paper is on longer-term forecasts?

Good point. The choices to use a binary decision and $\alpha$ =0.2 are illustrative to simplify the explanation of a complicated idea. Similar results were found when we conducted this experiment with different values of $\alpha$ and forecast lead-time, and additionally with multi-categorical decisions with different numbers of flow classes. This is noted on line 500-503 however we inadvertently forgot to mention forecast lead-times which will add to this sentence. We will also add an additional sentence on line 500 to clearly state that the chosen values (i.e. binary decision, $\alpha$ =0.2, 1st week of lead-time) were selected to simplify the explanation.

2.29. Line 510-511: is this a general conclusion? Over any lead time and situation? Not all probabilistic streamflow forecasts are skilful and reliable. Do you mean for the case study of the paper? Please, clarify.

Thanks for spotting this. We will make this sentence specifically about the forecasts used in this case study and add an additional sentence on the availability of streamflow post-processing methods to improve skill and reliability of raw forecasts.

2.30. Lines 513 and 514: I suggest using "developed" and "can be applied".

Well spotted. We will correct this.

2.31. Line 520: Please check deleting "is".

We will fix this typo.

2.32. Line 553: I do not understand what you mean by "a single forecast user" (single forecast or single user)? Please, clarify. Also "they" here refers to whom? The users?

Thank you for noting this. We are referring to a single decision-maker and will therefore replace the "forecast user" with "decision-maker". We will also replace "they" with "the decision-maker" so the sentence is clear.

2.33. Line 569: by "mitigation" do you mean "real time mitigation of damages"? Sometimes mitigation is more related to "prevention" (out of real time) for some users. Please, clarify.

Thanks for raising this important point. We will add an additional sentence explaining these two different types of "mitigation". In the context of a cost-loss economic model, mitigation refers to "preventive" action taken ahead of time. This is therefore what is meant by mitigation in REV and in our application of RUV. However, as RUV is general purpose and any economic model can be used, so it could in principle consider either of these types of mitigation. Exploration of this dynamic decision-making process over lead-times and forecast updates is left for future work and will be noted in section 6.3.

2.34. Section 6.3: I suggest using "could" instead of "will" when talking about possible future pathways for further research/future works.

Thank you. We will change this.

2.35. Overall: please check the use (or the absence) of a comma before the word "which".

Sure, we will check this.

2.36. Figures/tables: overall, please check the use of colours in black and white printing (maybe use italics in Table 3 instead of red, for instance; use dotted lines instead of colours in other figures, etc.)

Thank you for this suggestion. We will use italic and red in table 3 and will add additional line styles to improve ease of reading the figure. In addition, we will ensure the colour choices are colour blind friendly.

References McInerney, D., Thyer, M., Kavetski, D., Laugesen, R., Tuteja, N., & Kuczera, G. (2020). Multi-temporal Hydrological Residual Error Modeling for Seamless Subseasonal Streamflow Forecasting. *Water Resources Research*, *56*(11). https://doi.org/10.1029/2019wr026979

Murphy, A. H. (1977). The Value of Climatological, Categorical and Probabilistic Forecasts in the Cost-Loss Ratio Situation. *Monthly Weather Review*, *105*(7), 803-816. https://doi.org/10.1175/1520-0493(1977)105<0803:tvocca>2.0.co;2

Richardson, D. S. (2000). Skill and relative economic value of the ECMWF ensemble prediction system. *Quarterly Journal of the Royal Meteorological Society*, *126*(563), 649-667. https://doi.org/10.1002/qj.49712656313

---

## Author Response (AR1)

Dear Professor Buytaert,

We thank you for handling the editorial process for our manuscript, and the two reviewers for their constructive comments and suggestions.

We have taken on board the vast majority of the reviewers' suggestions and have revised the manuscript accordingly. The most notable improvements to our manuscript have come from:

1. Rewriting and reorganising the Abstract, Introduction and Conclusions based on comments from both Reviewer 1 and 2. These sections now largely focus on the introduction of RUV, and the outcomes of evaluating the sensitivity of RUV to aspects of decision context, which are the main contributions of our paper. See detailed response to comments 1.3, 1.26, 2.2, 2.3, 2.4, 2.6, 2.16, 2.19, and 2.26 below.
2. Removing ambiguity around some concepts by using consistent terms and clearly defining their meanings (based on comments from both Reviewer 1 and 2). This includes changing the title of Section 4 to "Illustrative case study". See detailed responses to comments 1.3, 1.5, 1.6, 1.10, 1.21, 2.8, 2.14, and 2.18 below.
3. Updating all figures for clarity, including an additional panel to Figure 3 showing the utility function for different levels of risk aversion, additional lines to Figures 4 and 5, and separate panels for each lead-time rather than decision-type in Figure 6. Interpretation of these additions was also added to relevant sections. See detailed responses to comments 1.17, 1.29, 1.30, 1.33, and 2.36 below.

In our detailed response, we respond to individual comments from the two reviewers. We have itemized all the comments for ease of reference, and our responses are in red text for your reading convenience.

We once again thank the reviewers for their many helpful comments and suggestions which have substantially improved the quality of our manuscript.

Yours sincerely,

Richard Laugesen and co-authors

2 December 2022

**Response to comments from Reviewer 1**

**Summary**

1.1.    The manuscript by Laugesen et al. introduces a new metric to assess forecast value adapting the formulation of a previously existing metric, namely Relative Economic Value, within a flexible value assessment framework based on utility. The method is then exemplified with subseasonal forecasts in the case of the Murray River, Australia, where decisions tend to target high flow values. A sensitivity analysis is carried in this case study.

The paper, which proposes a new methodology and results of high significance for the forecasting community, is detailed, very didactic and of high quality, and will undeniably be valuable to researchers who wish to carry out advanced and flexible forecast value analyses, involving decision-makers' levels of risk aversion.

I strongly recommend this paper for publication and list hereafter recommendations for clarification, as well as some minor points and typos.

We thank Reviewer 1 for their encouraging feedback and detailed review of our paper. In particular, we appreciate their thoughtful suggestions for improving the clarification of certain sections, which have made this material easier to follow.

**Comments**

1.2.    L18-21: These two sentences seem a bit contradicting because you first announce value for all lead times, decision types and most levels of risk aversion, but then you nuance your statement beyond the second week, for binary decisions. I suggest nuancing the first statement.

We agree that these two sentences appeared contradictory and have adjusted the abstract to rectify this and other issues.

1.3.    In addition, the case of the Murray-Darling basin being an example of application for sensitivity analysis rather than a stand-alone evaluation, I would consider these results as secondary compared to the advantages of the proposed RUV metric and the results of the sensitivity analysis well described in Section 6.2, which in themselves deserve to be highlighted in the abstract.

Good point. We agree that the Murray Darling experiments are an example of a "sensitivity analysis", more so than an evaluation of forecasts, and have revised the text to reflect this. In particular, we have modified the abstract to highlight the outcomes of the sensitivity analysis in section 6.2, used the term "illustrative case study" (L332), and shifted the emphasis in the abstract and conclusions from the stand-alone evaluation of forecast value to the outcomes of the sensitivity analysis.

We were concerned that the term "sensitivity analysis" is commonly used in a different context in the hydrological literature (e.g. exploring output response to uncertainty in parameters), and could cause confusion here. Therefore, we have avoided the use of "sensitivity analysis", and use the following wording to explain our approach (L17-20):

*"The key differences and similarities between REV and RUV are highlighted, with a set of experiments performed to explore the sensitivity of RUV to different decision contexts (binary, multi-categorical, and continuous-flow decisions, and various levels of user risk aversion and mitigation expensiveness)."*

1.4. L20: "Beyond the second week" please mention that you are referring the lead time.

Good suggestion. We have addressed this in the updated abstract (L27).

1.5. L26 (and throughout the paper): Here authors refer to the lens of "consumer" impact. The terms "user" and "decision-maker" are also used throughout the paper. Given that there are differences between these terms, I wonder whether the authors could clarify whether they use these three terms as interchangeable, or do they make a distinction. In the former case, are they actually interchangeable? Forecast datasets are increasingly open, and I am not sure whether users are indeed consumers in these cases. In the latter case, could you explicit the distinction made in an evaluation context?

This is a good point. We agree that these terms were not used consistently and have corrected this by using the term "user" throughout the paper. We feel it is important to be clear that the "user" in this context is an individual making a decision and therefore we have replaced "decision-makers" with "decision-makers (users)" - see line 11.

1.6. L73-78: Based on these two examples, and purely intuitively, I would tend to consider both types of decision-makers to be risk averse (conservative approach to avoid spending in example 1 and flooding in example 2) but with a different sensitivity to forecast uncertainty. Could the authors elaborate on why they make a direct link between forecast uncertainty and risk aversion?

Thank you for raising this important point of clarification. There is potential for confusion, as the formal definition of "risk aversion" does not always align with the colloquial interpretation used in daily life. In our study we used the term "risk aversion" from the economic literature, as defined on lines 73-74 (original version). However, we see that this definition was not entirely clear and have replaced it with the following definition on lines 86-87, "A user is said to be risk averse if they prefer an option with a more certain outcome, even if it may on average lead to a less economically beneficial outcome" with a citation to (Werner, 2008). We have also removed the second example from the paper, as we now appreciate that it is unnecessary and complicates the introduction of the "risk aversion" concept.

1.7. L95: Maybe reformulate "lead to improved forecast verification". For instance: "lead to improved forecast verification indicators" or "improved forecast performance".

Good suggestion. We have changed this wording to "improved forecast performance".

1.8. L98: "first convert them"

Thanks. We have corrected this.

1.9. L125: Isn't it a 2x2 contingency matrix?

Yes it is. We have fixed this typo.

1.10. L133: The term "outcome" was unclear to me here. I was unsure whether it referred to each combination of possible Action/Event in Table 1. In my understanding, E depends on each information source (reference, forecast, or perfect) but uses all possible outcomes in its weighted mean. The term "outcome" was a bit confusing, while Equation 1 and L138 were perfectly clear. Since the Supplement helped in that matter, I would suggest referring it here already.

Thanks. We have replaced the term "outcome" with "combination of action and occurrence" throughout the paper and supplement (e.g., L137).

1.11. Equation 1: Could you please add the range within which V should fall (-∞ to 1)?

Good suggestion. We have added the range.

1.12. Equation 2: At this stage o is not defined.

Thanks for catching this oversight. We have added a definition for $\bar{o}$ as "the frequency of the binary decision event" (L153) in the sentence following equation 2.

1.13. Figure 1: The location of the phrase "Use reference to decide" is, I think, misleading. Based on the explanations (L162-164), it seems that for a cost-loss ratio of 0.5, for instance, the forecast outperforms climatology and should thus be used to decide, with a potential REV reaching about 0.8. Therefore, using the reference for a cost-loss ratio of 0.5 would not

allow reaching a REV greater than that of the forecast. However, based on the figure, it seems that using the reference for a cost-loss ratio of 0.5 would allow reaching a REV greater than that of the forecast. Maybe the arrows pointing at the extreme intervals when the reference is indeed performing better, but this is currently not clear.

We see how the position of the "use reference to decide" text could introduce confusion.

We have replaced the annotations on the figure with clearly labelled regions of the value diagram and added concise text on how to interpret each.

(a) $0<=\alpha<0.05$ – Region 1: Baseline preferred for deciding action

(b) $0.05<=\alpha<0.95$ – Region 2: Forecast preferred for deciding action

(c) $0.95<=\alpha<1$ – Region 3: Baseline preferred for deciding action

Additionally, we removed the decision-making implications for region 1 and 3 when a fixed average climatology value is used as the baseline (i.e., "always act" and "never act" respectively). This makes this illustrative figure a more general introduction to a Value Diagram.

We believe that the revisions have made this figure much clearer and thank reviewer 1 for the suggestion.

1.14.   Additionally, it is not clear whether the arrows linked to "Always act" and "Never act" point at the interval when climatology < forecast or at the specific points (0;0) and (0;1) (see also the following response to comment 1.15).

Thanks for highlighting this. Our intention was for the arrows to refer to the intervals rather than (0,0) and (0,1). We have adjusted the figure to make this clearer as detailed in our response to comment 1.13.

1.15.   Figure 1 (and all value diagrams): If I understand correctly the meaning of $\alpha=1$ (never worth acting) and $\alpha=0$ (always worth acting), the decision can be taken regardless of whether the forecast or climatological information is considered. This would mean that the relative economic value should be exactly equal to 0 in both cases ($\alpha=1$ and $\alpha=0$). If that is correct, and that no other parameter comes into the decision of acting or not, is there a reason why the two points (0;0) and (0;1) are not represented in the value diagram?

Thanks for this comment. The above interpretation of the cases $\alpha=1$ and $\alpha=0$ is not quite correct. There is no conceptual reason why the relative value should be zero at these end points as this would imply that the forecast and reference baseline are both equally valuable, which is unlikely. In our illustrative example the reference baseline is more valuable.

REV uses a fixed average value for the reference baseline and there are only two possible actions:

1. Always act (when $\alpha < \bar{o}$ ) or
2. Never act (when $\alpha >= \bar{o}$ )

(where $\bar{o}$ is the observed event frequency).

This is detailed in the derivation on lines 30-34 of the supplement.

A decision-maker *should* prefer climatology to make decisions when climatology is more valuable than forecasts, and therefore they will use one of these two options if their $\alpha$ value lies in intervals the arrows are pointing at in our illustrative example.

We have added a reference to the derivation in the supplement, and cited (Richardson, 2000) which introduces REV, the value diagram, and its interpretation.

Additionally, we removed the "always act" and "never act" annotation as detailed in our response to comment 1.13.

1.16. Equation 4 (and Equation 9): Probabilities being sometimes used with powers, I would suggest to place the index m as a subscript rather than superscript.

Thanks for pointing this out. We now use subscripts (rather than superscripts) for the *m* index (used for probabilities) in all affected equations in the paper and supplement, e.g., equation 4 (L220).

1.17. L207-217: I suggest adding an example graph of μ to illustrate your explanation. For instance, I find it hard to picture the concavity of μ, especially in the case of binary decisions.

Thank you for this suggestion. We have added a second panel to figure 3 with a graph of $\mu$ for the 4 values of *A* used in the study. We have also add an explanatory sentence to the risk aversion section 4.5 (L390-391) and referenced the new panel in section 2.2 (L226). This is an excellent addition which makes the linkage between streamflow to damages to utility much clearer.

1.18. L230: "the absolute value of a specific decision"

Thank you for picking this up. We have addressed this typo while reworking this paragraph for clarity (L244-250).

1.19. Equation 6: Here it is not clear to me why damage does not vary with time (in Appendix A it seems it does). It is also not clear why m, whom E, b and d depend on, appear in parenthesis in the case of b and d, and as a subscript in the case of E.

We agree that there is inconsistency in our notation, especially with the way we are indexing *t* and *m*.

The reviewer is correct that observed damages vary with time, as the realised state of the world associated with each observation changes with time. We have made this clear in our revised notation for equation 6, across the rest of the paper and supplement (e.g., L252).

1.20. L237 "The damage function relates the streamflow magnitude to the economic damages": At this stage, you have not mentioned streamflow yet, I would suggest sticking to the term "states of the world".

We agree that we should refer to "states of the world" here, and have made this change (L257). Thanks!

1.21. L309: The previous section also comprised elements of methodology. Consider changing the name of this section.

Good suggestion. We have changed this section title to "Illustrative case study" as it more appropriately describes the content. This is further explained in our response to comment 1.3.

1.22. Section 4.2: Could you briefly state why you chose this station and basin?

Good idea. We have explained that this site is significant for water resource management because it is upstream of a major water storage (L348).

1.23. To which extent do you expect your results (sensitivities) to differ in a catchment with different hydrometeorological characteristics?

We have added a sentence to section 6 that the results are likely to be sensitive to the flow characteristics and forecast uncertainty, and that sites with other hydroclimatic conditions will be analysed in future work (L663-664).

1.24. L340-341: Given that you mention a rainfall post-processing step, I would recommend stating "raw streamflow forecasts" (L340) and "the streamflow observations" (L341) to avoid any misunderstanding.

Thanks, this will avoid potential confusion. We have made this change (L358-360).

1.25. Section 4.3: GR4J also uses temperature or potential evapotranspiration as input. Could you say something about what you used?

Good catch. We have added that PET from the AWAP model has been used, along with a citation (L358-360).

1.26. L345-346 "flow exceeding the height of a levee": it would be more intuitive to talk about the "water level exceeding the height of a levee"

We agree, and have made this change. Thanks. This section was incorporated into the introduction to address a comment from reviewer 2 (2.19), and is now located at L64.

1.27. L374: "all decision-makers share the same level of risk."

Thanks. We have fixed this typo (see L386).

1.28. Table 3: (1) "Experiment 4: Impact of risk aversion on forecast value"; (2) In experiment 5, the decision thresholds says "All flow" but the decision type is "Binary", which is counter-intuitive. "All possible thresholds" might be easier to understand, or "Thresholds from bottom 2% to top 0.04%".

Good suggestion. We have changed the text to reference the thresholds used.

1.29. Figure 4: Here you consider two rather extreme yet probably realistic thresholds for converting the probabilistic forecast into a deterministic one. When reading the results, I was wondering whether moderate thresholds could alleviate the lack of forecast value for high and low cost-loss ratios and provide reasonable value for all cost-loss ratios. Could you answer this by displaying intermediate probability thresholds in this experiment?

Thank you for suggesting this. We have added an additional intermediate probability threshold of $p_\tau = 0.5$ to further illustrate this issue.

1.30. Figure 6: To ease the reading of this figure whose lines are plain and with colors of similar intensity, I suggest adding dashes and dots to distinguish the three curves.

Good suggestion. We have added additional line styles to make it easier to read the figure and ensured the colour choices are colour blind friendly by using the colour palette proposed in Wong (2011).

1.31. Figure 6: Could the authors explain the interesting difference in RUV pattern for the multi-categorical decision (also seen in other decision types) between lead week 2 and lead weeks 3 and 4? Why does value decrease with lead time for low cost-loss ratios (as expected) but increases with lead time (maybe less obvious) for high cost-loss ratios?

Thank you for bringing this pattern to our attention. While the differences between weeks 2 and 3/4 are minor they interestingly appear robust to decision-type. We have added a short paragraph to section 5.3 noting this finding and providing a possible reason for the differences, namely "lead-time dependent differences in forecast reliability and decreasing sharpness of the forecast ensemble at longer lead-times" (L481-486). A definitive explanation would require a dedicated experiment and will be sought in future work that focuses on a decision-maker application and/or additional forecast locations. We have mentioned this future research direction explicitly in Section 6.3 on future work (L633-635).

1.32. Experiment 3: In this experiment, authors look at the variation of value with the lead week. It is also common to look at the influence of the initialization month or season to appreciate the influence of different hydrological conditions on the value. Even though it would mean dividing the total forecast sample into subgroups and reducing significance, I think it could be a valuable addition to Figure 6 to show forecasts initialized in dry and wet conditions separately.

Yes, this is a great point and we agree that assessing the impact of seasonality and antecedent conditions on forecast value is an important research question. While important, we believe this assessment does not fit is not within the research aims of the current paper, which focus on the introduction of RUV and a comparison to REV.

The Biggara case study is used as a vehicle to illustrate the general application of the RUV method. The current set of case study analyses is already quite extensive and includes results from 5 experiments illustrated through 8 figures. To sufficiently assess the impact of different hydrological conditions on forecast value we would need to add an additional experiment with dedicated figures and text to explain the impact of the seasonality and antecedent conditions with respect to the decision-types and lead-times. Further, for completeness the impact should also be assessed on the risk aversion outcomes through additional evaluation in experiments 4 and 5. We feel that this additional content would distract from the main aims of the paper and result in an unduly lengthy manuscript.

After some reflection, we feel this interesting and important research question requires a dedicated separate study and is best left for future work. Thank you for raising this important issue for practical application. We have added text to Section 6.3 that addresses this and outlines further work is needed (L668-670-673).

1.33. Figure 7: To ease reading, consider adding a horizontal line at y=0 in graphs displaying the overspend.

This is a good suggestion. We have done this.

1.34. Experiment 4: It is currently unclear why the third line of Figure 7 is shown as it is little to not exploited in the interpretation. Please consider removing or spending some sentences to exploit this line of the figure.

Thank you for this suggestion. We assume you are referring to the third row of panels rather than third line. One intent of including the utility-difference (and overspend) results was to enable a more direct comparison of our findings with those in Matte et al. (2017), line 469-471. We agree that adding an interpretation of the utility-difference results is important, and have added this on L507-510.

1.35. L520: "making decisions with fixed critical probability thresholds leads to"

Good catch. We have fixed this typo.

1.36. Sections 6.1, 6.2 and 6.3: Numbering the paragraphs is unnecessary.

Thank you for highlighting this. We have removed the paragraph numbering from these sections.

1.37. L576: "summarizes"/"summarises"

We have corrected this Australian/US language issue, and checked the rest of the document.

1.38. L680 and Table 5: In the text, you mention that the formulation of Ct depends on the value of p, but in Table 5, the formulation of Ct depends on whether the action is taken or not rather than on p. I could not figure out why. Are the p values you are referring to in both instances different? Could you please clarify this point?

Thank you for pointing this out. We agree that it was not clear from the text in the appendix. Section 2 in the supplement included a more complete derivation but on reflection that was also lacking clarity.

The probability on line 680 was referring to the forecast probability of flow above the threshold, which then determines ex ante (i.e. before event has taken place) how much to spend on the action $\bar{C}_t$ . The probability in Table 5 was referring to the ex post probability that the observed flow is above the threshold. The amounts spent on action or no action in the row titles of Table 5 are the optimal costs $\bar{C}_t$ found ex ante (i.e. after event has taken place).

We have made the following changes to the main text and the supplement to improve the clarity of this derivation:

- On line 735 replaced "probability is always 1 or 0" with "forecast probability is always 1 or 0"
- Changed the column titles of Table 4 and Table S2 in the supplement to "Event forecast to occur" and "Event forecast to not occur".
- Changed the column titles of Table 5 and Table S3 in the supplement to "Event occurred" and "Event did not occur".
- On line 728 replaced "forecast probability is 1 or 0" with "event is forecast to occur (p=1) or not occur (p=0).
- On line 118 of the supplement replaced "letting the probability be conditioned on observed flow above the threshold" with "letting the probability be conditioned on observed flow above the threshold, rather than the forecast flow used for the ex ante utility".

1.39. L701: The link to the companion dataset is missing.

Thanks for pointing this out. We have included a permanent link to the companion dataset in the revised manuscript.

**Response to comments from Reviewer 2**

**Summary**

2.1. The paper presents a generalisation of the Relative Economic Value (REV) approach, providing a flexible metric, the "Relative Utility Value" (RUV), which can be useful for decision makers on the value of probabilistic subseasonal forecasts. The results show its application and sensitivity to several factors in a case study in Australia.

The paper is well written and demonstrated. I believe it brings novel aspects in the topic of hydrometeorological forecasting, and is an excellent demonstration of how forecast producers and users should work together to enhance the usefulness of skilful forecasts.

I have just some minor general and specific comments, presented below.

*Thank you for this thorough, well considered, and detailed review. We appreciate your words of encouragement and suggestions which have improved the quality of this work.*

**General comments:**

2.2. I think some sentences need to be more carefully revised because they might convey a message that goes beyond the experimentations of this paper. For instance, concerning the first sentence of the Conclusions section, I do not believe that, overall, the value of probabilistic forecasts to making (good) decisions has not been established, as the authors say. Many public and private companies are convinced of the value of quantifying uncertainties in real-time forecasting and that is why this type of forecasts has been increasingly produced and used for many operations, from nowcasting to short-term flood forecasting and long-term inflows to reservoirs. Value has not been established (or explicitly calculated) at all lead times and users cases, I agree, but, overall, the forecasting (producers and users) community acknowledges that there is value for decision making in not being certain (or deterministic) about the unknown future. The added value of the paper, in my opinion, does not lie on bringing the "value" into discussion in forecast verification/evaluation, as this has been done in several papers previously, but in making the framework for assessing it more accessible and flexible, as the title says.

*Thank you for bringing this important point to our attention. We now recognise that some statements in our paper were too general and went beyond the experiments conducted in our paper. In particular, we have extensively rewritten parts of the Conclusions, Abstract, and Discussion which has addressed this problem.*

*For example, in the Conclusions we removed "A case study demonstrates that subseasonal streamflow forecasts should be preferred over a reference climatology forecast for all lead-times studied (max 30 days) and almost all decision-makers regardless of their risk aversion" with "An illustrative case study using probabilistic subseasonal streamflow forecasts in a practically significant catchment in the Southern Murray-Darling Basin of Australia was used to compare the REV and RUV metrics under a range of decision contexts" followed by an itemised list of key findings specific to this case study, e.g., "2.*

Forecast value depends on the decision type and hence, it can be critically important to use a decision-type that matches the real-world decision".

Our changes made in response to the next comment (point 2.3) have further aligned the message with the experimental results.

We also agree that the main contribution of this study is to introduce a more flexible framework for quantifying forecast value, rather than to establish the value of probabilistic forecasts in all cases, or their value over deterministic forecasts. Although this was stated in the research aims and supported in the Introduction, it was not clear that this is the main contribution. We have rewritten the Abstract and Conclusions to ensure this main contribution is more evident.

2.3.  I was also puzzled by the authors when they say that a decision maker who is highly exposed to damages should use the reference climatology rather than a forecast based on meteorological numerical models for binary decisions (Conclusions, lines 639-640). This might be the case for the experiment showed (and the case described in the paper), but I doubt flood forecasters (forecasting a threshold exceedance for the next 12-24 hours, for instance) would be able to say to the population they are serving that they will abandon a city located close to a river and leave than with only a climatology-based information instead of rather investing into a (good) model-based forecasting and alert system because they are highly exposed to damages. I fully understand that if the potential costs of a flood event are high, and will be incurred if the flood occurs, whatever forecast we might deliver, then no forecasting system can save us, and it is better to work on protection (decreasing costs) at first. But even in this case, using climatology might not be beneficial either (the problem is elsewhere, not in the type of forecast being used). What I mean is that out of a more explicitly presented context, some sentences might rather diverge a reader from the purposes of the paper. Therefore, I would recommend to revise some general affirmative sentences, or at least introduce more context to them to avoid misunderstandings.

Thank you for this considered comment. Reviewer 1 (point 1.3) also asked that we revise the conclusions to focus more on the outcomes of the experiments that explore the sensitivity of RUV to different decision contexts, rather than outcomes of the case study. We have extensively rewritten the Conclusions and Abstract, and have removed overly general affirmative sentences. We have also stated that the outcomes are context dependent and provided clear information on that context.

2.4.  a) Another general comment is about the fact that we set the context of the paper on probabilistic subseasonal forecasts (up to 30 days), but much of the demonstrations and experiments refer to 1-7 day lead-time forecasts (and many concluding sentences seem to forget this context and generalize to any type of forecast and lead time).

Thank you for highlighting this. During preliminary investigations we did generate results for other lead-times and groupings. The specific case-study features used (as described in section 4) were selected to best present the salient features of RUV and the case study results. This is described on lines 333-337. On reflection, we agree with the reviewer that some concluding statements were generalized beyond the experimental results shown in Section 5. We have rectified this while extensively rewriting the conclusions. Please see related responses to comments 2.2 and 2.3 and L680-697.

b) In many situations (but I am not sure about the case of the particular catchment of the study), a meteorological (model-based) forecast may show quality a couple of days ahead (1 to 5 days, for instance) and then be as skilful as climatology afterwards. How this difference in the quality of the forecasts might affect the results here? Is it justified to group together these lead times here?

This is an interesting point. We agree that rainfall forecasts are only skilful for short lead times (e.g. 1-5 days). However, streamflow forecasts are typically skilful at longer lead-times than rainfall forecasts due to storage effects. In particular, the subseasonal streamflow forecasts used in this case study have previously been shown to be sharper and had higher CRPS skill scores than climatology for lead times up to 30 days (see McInerney et al. (2020) for a detailed explanation and evaluation of this. Additionally, Figure 6 demonstrates that these forecasts have higher value than the reference climatology baseline for longer lead-times.

We found that grouping lead times together in our analysis was beneficial in addressing the aims of this paper, namely to introduce RUV and contrast it with REV. We also found that the particular groups (1 week, 2 weeks, 3-4 weeks) demonstrated key differences between lead times.

However, grouping lead-times would not be recommended in a practical context. If the purpose of this case study was to quantify value of using forecasts to inform river operations, then we would need to analyse forecast value of individual lead-times. We have added the additional sentence to L401; "Grouping lead-times together simplifies the introduction of RUV and comparison of its salient features with REV; however, for practical applications there may be benefits for evaluating forecast value at specific lead-times of interest." Please also see our response to a related comment (1.31) from reviewer 1.

c) Would a (potential) difference in quality explain negative RUV (lines 412-414), where the authors say that climatology (as a forecast) is more useful than a (meteorological model based) forecast?

We do not believe that the differences in forecast performance across the 1st week are the root cause of negative value regions discussed on line 428-429 of the original manuscript. The negative regions are a consequence of using a fixed critical probability threshold, as explained in section 5.1 and in the REV literature, see cited references Richardson (2000)

and Murphy (1977). This result is seen for any forecast regardless of whether lead-times are grouped or not. We have added an additional sentence at line 434 stating the generality of this result with reference to the REV literature.

d) (note: at the end, the decision maker is always using a forecast, either from a record of historic observations – climatology – or from a coupled atmospheric-hydrologic model).

This is of course true. However typically the term "forecast" refers to a procedure more complex than "just" the marginal distribution of historical data.

Thank you for bringing this need for more clarification to our attention. We agree that the term "forecast" typically refers to a reasonably complex procedure. As this study involves a comparison of the new method RUV to the existing method REV we are limited to using the baseline reference used by the REV. This critical limitation of REV is stated on lines 79-80 of the Introduction and is a key motivation for the development of RUV. We have included the need to apply RUV with "practically relevant reference forecasts" in Future work section 6.3 with an additional sentence at L640; "While the reference baseline (fixed average climatology) used in this study enabled a direct comparison of RUV with REV, we would recommend comparison against more relevant baseline forecasts for practical applications (e.g., information currently used to inform the decision being assessed)". Please see our response to a related comment (1.32) from reviewer 1.

2.5.    Finally, a last overall comment I have is: why a systematic comparison with REV is so important in the development of a novel approach or metric in this topic? Is it because REV is widely used (or supposedly widely used)? How crucial is it as motivation for the study?

The reviewer is correct that the comparison with REV is important because REV is commonly used. We feel this is an important motivation for the study, and make the following points about this in our original manuscript:

- REV is widely used in the literature for quantifying the value of forecasts (see lines 54-61),

- The primary way to present this information is using a value diagram (see lines 89-90).

- As the value diagram is a compelling way to communicate value across a range of decision-makers, and the community is familiar with it, we felt it was important to be able to leverage this with any new approach (see lines 265-266).

**Specific comments:**
2.6.    Introduction: I think the authors could introduce some literature on works done on forecast value and links between forecast quality and value with respect to inflows to

hydropower reservoirs. These cover a large range of cases and lead times, and also use optimisation-based economic models to link forecast production (quality) to usefulness (economic value). It would be interesting to give here this broader view to the topic, I think, and then replace better the context of the paper (to which the conclusions drawn will specifically apply). Besides the paper mentioned in the discussion (Penuela et al.), some others that might be interesting are: https://doi.org/10.1002/2015WR017864; https://hess.copernicus.org/articles/23/2735/2019/; https://doi.org/10.1029/2019WR025280; https://hess.copernicus.org/articles/25/1033/2021/.

Thank you for this suggestion and for providing these references. We have included a sentence on L59-62 highlighting the links between forecast performance and value, with mention of active research in hydropower modelling.

2.7.   Line 49: too many "and" words. Please, check.

We do not see any "and" words on line 49 of the original manuscript and feel the sentence reads fine.

2.8.   Line 50: "better verification implies more value": I think you refer to "quality" and not "verification". Please, check.

We agree that "verification" was the wrong term here. We have replaced "better verification implies more value" with "better forecast performance (according to our verification metrics) implies more value". Thanks for catching this.

2.9.   Line 88-89: not clear to me. Please, check.

Thanks. We have rewritten this sentence for clarity.

2.10.  Line 90, 102: when you refer to "the authors" I am sometimes a bit confused if you mean "you" or the authors in Matte et al. Please, check.

Thanks for noting this. We have replaced "the authors" with "we" in these two sentences.

2.11. Line 192-193: maybe it is not reported in scientific papers, but are you sure it is not commonly used by water managers in practice? Have you conducted a survey or any other study not reported here to assess it (i.e., real-world practices)?

Good point. This statement was based on our knowledge of the scientific literature and 15 years professional experience providing forecasts to water managers in Australia. However, we recognise that this statement was too broad, and have adjusted the language accordingly. In particular, we have replaced the sentence on lines 192-193 of the original manuscript with "To the best of our knowledge, studies of real-world decisions using this alternative approach ( $p_\tau = \alpha$ ) have not been reported in the published literature" (see lines 209-211).

2.12. Line 227-230: again too many "and" words. I found the sentence unclear. Please, check (maybe also correct to "a specific decision").

Thanks for spotting this. We have removed the unnecessary "and" from line 227 of the original manuscript. We assume that you are referring to the sentence on lines 228-230 as being unclear. We agree and have split this into two sentences, as well as added a little more context (see lines 246-250 of revised manuscript).

2.13. Line 280: I am not fully convinced that information on amount spent, damage etc. at each time step is something valuable to a user. Is that so? Can you provide examples or a justification for that? I believe that users might be more interested in the long-term performance of a forecast system (in particular when it comes to reservoir operations), while a flood alert user would be interested in the whole flood event duration performance (and less on each time step). Maybe I misunderstood something here.

Thank you for highlighting this. We have provided an example of users who may find this additional information useful with the following additional text at lines 300-304:

"*This may benefit a user applying alternative economic models and tuning damage functions to match real-world data, as they would require the amount spent and damages incurred at individual time steps to determine the components are behaving as expected. Additionally, a user who has finite funds to spend on mitigation and wants to determine when their budget will be exhausted would require investigation of spend and damage amounts at individual time-steps.*"

2.14. Line 309: I do not think "Methodology" is a good title for the section. I would suggest "Application" or "Experiment".

We agree with this suggestion, and have changed the heading of Section 4 to "Illustrative case study" to better describe the content.

2.15. Line 310-311: I guess that by "different decision-makers" you mean "different levels of aversion of decision-makers". I think it is not the person themselves you are talking about but the theoretical level of aversion that you are modifying in the experiments.

Correct. We are referring to the level of risk aversion of an individual decision-maker, and their exposure to damages ($\alpha$). We have adjusted "decision-makers" to "users with different relative expense of mitigation and different levels of risk aversion" on L335. Thank you for bringing this to our attention.

2.16. Section 4.1: I think part of it could go to the Introduction.

Thank you for this suggestion. We agree that some of this material could be included in the Introduction, and have moved lines 315-320 from the original manuscript to the first paragraph of the Introduction (see lines 33-40 of the revised manuscript).

2.17. Line 339: maybe place the references in the right place would help the reader (ex. Perrin et al., after GR4J, and not after RRP-S).

Thank you for this suggestion. We have moved the references from the end of the sentence and placed them after each modelling component.

2.18. Line 343: "seamless" has usually another meaning in the literature. It usually refers to a system that forecasts in a coherent and homogeneous way from minutes to hours and months. It is not usually related to performance across scales. Please, check.

In this context, "seamless" refers to forecasts which are coherent and homogenous and have similar forecast quality across time scales. We have clarified how we are using the term by rephrasing the sentence on L360-363:

*"The MuTHRE model was chosen for post-processing because it provides "seamless" forecasts that are (statistically) reliable and sharp across multiple lead-times (0-30 days) and aggregation time scales (daily to monthly). Further information on the forecasts used in this study can be found in McInerney et al. (2020), and further method improvements to enhance seamless performance in McInerney et al. (2021)."*

2.19.  Section 4.4: I think part of it could go to the Introduction (lines 346-354).

Thank you for this suggestion. We agree that the Introduction did not provide enough context or motivation for the different decision types and have included additional material to the Introduction based on the content in Section 4.4 (lines 346-354 of original manuscript).

2.20.  Line 369: what do you mean by "suitable"? How? Based on data?

Good point. We did not define what we meant by "suitable". We have clarified that the selected parameters reproduce the assumptions described in the previous sentence (see lines 379).

2.21.  Table 3, experiment 4: check typo

Well spotted. We have addressed the typo.

2.22.  Fig. 4: I am not sure it is needed to show that we come up to the same results.

Although the value diagrams in panel (a) and (b) are identical, they show results from two different methods. We feel it is important to show this equivalence in an explicit way as it supports the research aims of this study. We have changed the caption of panel (b) to "RUV with restrictive assumptions equivalent to REV" to make the difference between methods clearer.

2.23.  I would suggest putting Experiment 1 and Experiment 2 together.

Thank you for this suggestion.

Although the results shown in Figures 4 and 5 are similar, they are from different experiments with different reasoning and explanations. We prefer to keep the two experiments distinct for clarity and introduce the content in a staged way. We agree that combining the results together as a single figure would be more efficient in terms of space; however, it would be more difficult to explain and describe the differences. By keeping the results separate it is easier for the reader to understand the reasons for the similarity and differences between REV and RUV, which is a research aim of the paper.

2.24.  Line 437: what do you mean by "ensemble sampling error"? Please, explain.

Thank you for noting this. We were referring to errors due to the small ensemble size. We have replaced "ensemble sampling error" with "sampling errors from the relatively small ensemble size" on L455.

2.25. Line 458: please, clarify the sentence (see my general comments above) in terms of saying that a "decision-maker should avoid using forecasts" in certain conditions.

Thanks for pointing this out. We have replaced this statement with "This suggests the users should prefer the reference baseline for the binary decision and prefer forecasts for the multi-categorical and continuous-flow decisions. This highlights the importance of calculating forecast value using the decision type which matches the decision being assessed", L478-480. This point is discussed in our response to reviewer comment 2.3.

2.26. Line 464-466: Does this correspond to reality? Have you discussed the results with the Murray-Darling Basin managers, for instance? It would be interesting to link mathematical calculations to reality in the field, providing supporting to some sentences on the results and overall conclusions drawn in the paper.

We agree that it would be interesting to make a strong link between forecast value methods such as RUV and applications "in the field".

As pointed out by reviewer 1 (see comment 1.3), this paper is not aimed at providing a detailed and conclusive evaluation of forecast value for Murray Darling Basin managers. It is aimed at introducing RUV and contrasting it to REV by demonstrating that RUV can handle factors which are important to real-world decision making. This is demonstrated through experiments showing the conditions under which RUV and REV are equivalent and illustrating the sensitivity of forecast value to different decision types (binary, multi-categorical, and continuous flow) and levels of risk aversion. For the case study in this paper, we have made reasonable assumptions about the damage functions and decision thresholds, and necessarily used the cost-loss economic model for comparison with REV. We feel this is sufficient for the purposes of this paper, which is to demonstrate that forecast value is sensitive to decision-type and levels of risk aversion.

Now that we have established in this paper that forecast value is sensitive to these choices, in future work we can undertake a realistic evaluation of forecast value for Murray Darling Basin managers using RUV. We have added this topic to the future work section (L636-639) with the following text:

"*For practical applications of RUV it is advisable to calibrate the damage function, decision thresholds, economic model, decision-making approach, and reference baseline to the real-world experience of the decision-makers. This calibration will ensure the resulting forecast value is tailored to the specific decision context and will likely lead to more user trust in the results, and subsequently more appropriate use of forecast information.*"

2.27. Fig. 7: I think it should be more commented. The differences we see in the column on the right do not seem to be "moderate".

We have expanded our commentary of Fig 7 to ensure it is clear that the impact is "moderate", except for the case of highly risk aversion decision-makers with continuous flow (see lines 492-495 and 501-503). Thanks for pointing this out.

2.28. Experiment 5: could you justify the choice of adopting a binary decision and alpha = 0.2 here? Also, why are you showing week 1 if the focus of the paper is on longer-term forecasts?

Good point. Using a binary decision and $\alpha$ =0.2 provides an illustrative example for the introduction of a complicated idea. Similar results were found when we conducted this experiment with different values of $\alpha$ and forecast lead-time, and additionally with multi-categorical decisions with different numbers of flow classes (results not shown). This was noted on line 500-503 of the original manuscript; however we inadvertently forgot to mention forecast lead-times and have added these to this sentence at 521-523 in the revised manuscript. We have also included an additional sentence on L523-524; "The specific experimental values (binary decision, $\alpha$ =0.2, 1st week lead-time) were chosen as a representative example and the findings apply for other experimental values.".

2.29. Line 510-511: is this a general conclusion? Over any lead time and situation? Not all probabilistic streamflow forecasts are skilful and reliable. Do you mean for the case study of the paper? Please, clarify.

Thanks for spotting this. We have made this sentence specifically about the forecasts used in the case study and added an additional sentence on the availability of streamflow post-processing methods to improve skill and reliability of raw forecasts (L548-541).

2.30. Lines 513 and 514: I suggest using "developed" and "can be applied".

Well spotted. We have addressed this while rewriting the Discussion.

2.31. Line 520: Please check deleting "is".

We have addressed this typo.

2.32. Line 553: I do not understand what you mean by "a single forecast user" (single forecast or single user)? Please, clarify. Also "they" here refers to whom? The users?

Thank you for noting this. We are referring to a single decision-maker and have therefore replaced the "a single forecast user" with "a single decision-maker". We have also replaced "they" with "the decision-maker" so the sentence is clear.

2.33. Line 569: by "mitigation" do you mean "real time mitigation of damages"? Sometimes mitigation is more related to "prevention" (out of real time) for some users. Please, clarify.

Thanks for raising this important point. We have clarified the sentence by adding "preventative" to L599. In the context of a cost-loss economic model, mitigation refers to "preventive" action taken ahead of time. This is therefore what is meant by mitigation in REV and in our application of RUV. However, as RUV is general purpose and any economic model can be used, so it could in principle consider either of these types of mitigation.

Exploration of this dynamic decision-making process over lead-times and forecast updates is left for future work and has been noted in section 6.3 on L648-651; "Additionally, the cost-loss economic model used in this study implies that mitigation is preventative action to minimise forecast losses, with each forecast lead-time and forecast update treated independently of all others. Alternative economic models and decision-making frameworks may be required to explore more realistic forms of mitigation which consider temporal dependence"

2.34. Section 6.3: I suggest using "could" instead of "will" when talking about possible future pathways for further research/future works.

Thank you. We have changed this.

2.35. Overall: please check the use (or the absence) of a comma before the word "which".

We have checked this and addressed several uses. Thanks.

2.36. Figures/tables: overall, please check the use of colours in black and white printing (maybe use italics in Table 3 instead of red, for instance; use dotted lines instead of colours in other figures, etc.)

Thank you for this suggestion. We have now use italic and red in table 3 and have added additional line styles to improve ease of reading the figures. In addition, we have ensured

the colour choices are colour blind friendly by using the colour palette proposed in Wong (2011).

**References**

Matte, S., Boucher, M.-A., Boucher, V., & Fortier Filion, T.-C. (2017). Moving beyond the cost–loss ratio: economic assessment of streamflow forecasts for a risk-averse decision maker. *Hydrology and Earth System Sciences*, *21*(6), 2967-2986. https://doi.org/10.5194/hess-21-2967-2017

McInerney, D., Thyer, M., Kavetski, D., Laugesen, R., Tuteja, N., & Kuczera, G. (2020). Multi-temporal Hydrological Residual Error Modeling for Seamless Subseasonal Streamflow Forecasting. *Water Resources Research*, *56*(11). https://doi.org/10.1029/2019wr026979

McInerney, D., Thyer, M., Kavetski, D., Laugesen, R., Woldemeskel, F., Tuteja, N., & Kuczera, G. (2021). Improving the Reliability of Sub-Seasonal Forecasts of High and Low Flows by Using a Flow-Dependent Nonparametric Model. *Water Resources Research*, *57*(11). https://doi.org/10.1029/2020wr029317

Murphy, A. H. (1977). The Value of Climatological, Categorical and Probabilistic Forecasts in the Cost-Loss Ratio Situation. *Monthly Weather Review*, *105*(7), 803-816. https://doi.org/10.1175/1520-0493(1977)105<0803:tvocca>2.0.co;2

Richardson, D. S. (2000). Skill and relative economic value of the ECMWF ensemble prediction system. *Quarterly Journal of the Royal Meteorological Society*, *126*(563), 649-667. https://doi.org/10.1002/qj.49712656313

Werner, J. (2008). Risk Aversion. In (pp. 1-6). Palgrave Macmillan UK. https://doi.org/10.1057/978-1-349-95121-5_2741-1

Wong, B. (2011). Points of view: Color blindness. *Nature Methods*, *8*(6), 441-441. https://doi.org/10.1038/nmeth.1618